# What training reveals about neural network complexity

**Andreas Loukas**
EPFL
andreas.loukas@epfl.ch

**Marinos Poiitis**
Aristotle University of Thessaloniki
mpoiitis@csd.auth.gr

**Stefanie Jegelka**
MIT
stefje@mit.edu

## Abstract

This work explores the Benevolent Training Hypothesis (BTH) which argues that the complexity of the function a deep neural network (NN) is learning can be deduced by its training dynamics. Our analysis provides evidence for BTH by relating the NN's Lipschitz constant at different regions of the input space with the behavior of the stochastic training procedure. We first observe that the Lipschitz constant *close to the training data* affects various aspects of the parameter trajectory, with more complex networks having a longer trajectory, bigger variance, and often veering further from their initialization. We then show that NNs whose 1st layer bias is trained more steadily (i.e., slowly and with little variation) have bounded complexity even in regions of the input space that are *far from any training point*. Finally, we find that steady training with Dropout implies a training- and data-dependent generalization bound that grows *poly-logarithmically* with the number of parameters. Overall, our results support the intuition that good training behavior can be a useful bias towards good generalization.

## 1 Introduction

Though neural networks (NNs) trained on relatively small datasets can generalize well, when employing them on unfamiliar tasks significant trial and error may be needed to select an architecture that does not overfit [1]. Could it be possible that NN designers favor architectures that can be easily trained and this biases them towards models with better generalization?

In the heart of this question lies what we refer to as the "*Benevolent Training Hypothesis*" (BTH), which argues that *the behavior of the training procedure can be used as an indicator of the complexity of the function a NN is learning*. Some empirical evidence for BTH already exists: (a) It has been observed that the training is becoming more tedious for high frequency directions in the input space [2] and that low frequencies are learned first [3]. (b) Training also slows down the more images/labels are corrupted [4], e.g., the Inception [5] architecture is $3.5\times$ slower to train when used to predict random labels than real ones. (c) Finally, Arpit et al. [6] noticed that the loss is more sensitive with respect to specific training points when the network is memorizing data and that training slows down faster as the NN size decreases when the data contain noise.

From the theory side, it is known that the training of shallow networks converges faster for more separable classes [7] and slower when fitting random labels [8]. In addition, the stability [9] of stochastic gradient descent (SGD) implies that (under assumptions) NNs that can be trained with a small number of iterations provably generalize [10, 11]. Intuitively, since each gradient update conveys limited information, a NN that sees each training point few times (typically one or two) will not learn enough about the training set to overfit. Despite the elegance of this claim, the provided explanation does not necessarily account for what is observed in practice, where NNs trained for thousands of epochs can generalize even without rapidly decaying learning rates.

35th Conference on Neural Information Processing Systems (NeurIPS 2021).

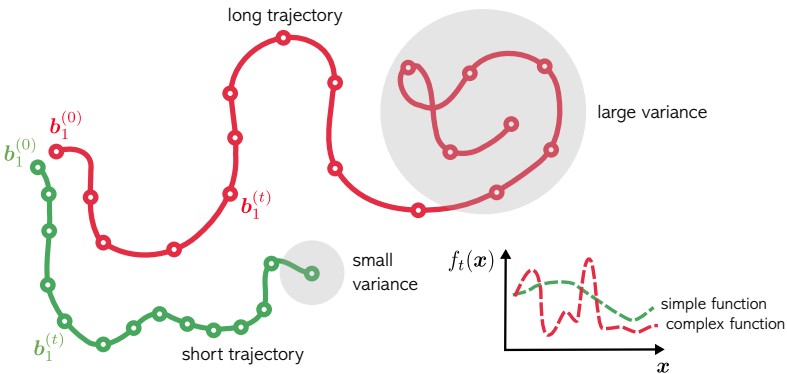

Figure 1: Our findings connect training dynamics and NN complexity by showing that the trajectory of the 1st layer bias reflects the NN's Lipschitz constant near (and far from) the training data: the bias of higher complexity NNs exhibits a longer trajectory and varies more at the end of the training.

## 1.1 Quantifying NN complexity

This work takes a further step towards theoretically grounding the BTH by characterizing the relationship between the SGD trajectory and the complexity of the learned function. We study neural networks with ReLU activations, i.e., parametric piece-wise linear functions. Though many works measure the complexity of these networks via their maximum number of linear regions [12–15], it is suspected that the average NN behavior is far from the extremal constructions usually employed theoretically [16].

We instead focus on the Lipschitz continuity of a NN at different regions of its input. For networks equipped with ReLU activations, the Lipschitz constant in a region is simply the norm of the gradient at any point within it. The distribution of Lipschitz constants presents a natural way to quantify the complexity of NNs. Crucially, NNs with a bounded Lipschitz constant can generalize beyond the training data, a phenomenon that has been demonstrated both theoretically [17–19] and empirically [20]. The generalization bounds in question grow with the Lipschitz constant and the intrinsic dimensionality of the data manifold, but not necessarily with the number of parameters[1], which renders them ideal for the study of overparameterized networks.

## 1.2 Main findings: connecting training behavior and neural network complexity

We link training dynamics and NN complexity close and far from the training data (see Figure 1).

**NN complexity close to the training data.** Section 4 commences with a simple observation: SGD updates the 1st layer bias more quickly if the learned function has a large Lipschitz constant near a sampled data point. This implies that the length of the bias trajectory grows linearly with the Lipschitz constant of the NN on its linear regions that contain training data (Theorem 1). Based on this insight, we deduce that (a) near convergence, the parameters of more complex NNs vary more across successive SGD iterations (Corollary 2), and (b) the distance of the trained network to initialization is small if the learned NN has a low complexity (near training data) throughout its training, with the first few high-error epochs playing a dominant role (Corollary 3).

**NN complexity far from the training data.** Section 5 focuses on the relationship between training and the Lipschitz constant in empty regions of the input space, i.e., linear regions of the NN that do not contain training points. We first show that the Lipschitz constants in empty regions are linked with those of regions containing training points (Theorem 2). Our analysis implies that NNs whose parameters are updated more slowly during training have bounded complexity in a larger portion of the input space. We then demonstrate how training NNs with Dropout enables us to grasp more information about the properties of the learned function and, as such, to yield tighter estimates for the global Lipschitz constant. Our findings yield a *data-* and *training-dependent* generalization bound

---

[1]While the Lipschitz constant is typically upper bounded by the product of spectral norms of the layer weight matrices (thus yielding an exponential dependency on the depth), the product-of-norms bound is known to be far from the real Lipschitz constant [21, 22]

that features a *poly-logarithmic* dependence on the number of parameters and depth (Theorem 3). On the contrary, in typical NN generalization bounds the number of samples needs to grow nearly linearly with the number of parameters [23–25] or exponentially with depth [26–30].

All proofs can be found in Appendix B, whereas Appendices A and C contain additional empirical and theoretical results, respectively.

## 2   Related works

**The Lipschitz constant of NNs.** Since exactly computing the Lipschitz constant is NP-hard [31], its efficient estimation is an active topic of research [31–34, 22, 35]. Our work stands out from these works both in motivation (i.e., we connect training behavior with NN complexity) and in the techniques developed (we are not employing any complex algorithmic machinery to estimate the Lipschitz constant of a trained model, but we bound it as the NN is being trained based on how weights change). Empirically, Lipschitz regularization has been used to bias training towards simple and adversarially robust networks [36–42]. Theoretically, the Lipschitz constant is featured prominently in the generalization analysis of NNs (e.g., [26–28]), but most analyses depend on sensitivity w.r.t. parameter perturbation, which is related but not identical to the Lipschitz constant.

**Dropout and generalization.** The Dropout mechanism and its variants are standard tools of the NN toolkit [43–45] that regularize training [46, 47] and help prevent memorization [6]. The effect of Dropout on generalization have been theoretically studied primarily for shallow networks [48, 49, 47] as well as for general classifiers [50]. The generalization bounds that apply to deep networks are norm-based and generally grow exponentially with depth [51, 52] or are shown to scale the Rademacher complexity by the Dropout probability (for Dropout used in the last layer) [53]. We instead base our analysis on arguments from [18, 19] and exploit the properties of ReLU networks to derive a bound that features a significantly milder dependency on the NN depth.

**Flat and sharp minima.** Flat minima correspond to large connected regions with low empirical error in weight space and have been argued to correspond to networks of low complexity and good generalization [54]. It has also been shown that SGD converges more frequently to flat minima [55–59]. Different from the current work that focuses on the sensitivity w.r.t. changes in the data, flatness corresponds to a statement about local Lipschitz continuity w.r.t. weight changes. In addition, whereas flat minima are regions of the space where the loss is low, our main results account for more complex loss behaviors (by means of an appropriate normalization). Note also that some works argue that the flat/sharp dichotomy may not capture all necessary properties [60–62] as flat minima can be made sharp by a suitable reparameterization [61], and flat and sharp minima may be connected [61].

**Training dynamics of NNs.** Many authors have studied the training dynamics of NNs [63–69], arguing that, with correct initialization and significant overparameterization, SGD converges to a good solution that generalizes. Our work complements these studies by focusing on how the SGD trajectory can be used to infer NN complexity. Arora et al. [8] connect the trajectory length and generalization performance via the Neural Tangent Kernel (NTK). Most analyses based on the NTK ("lazy" regime) or mean field approximation ("adaptive" regime) focus on 2- or 3-layer networks. In contrast to these works, we make no assumptions on initialization or NN size.

## 3   Preliminaries and background

Suppose that we are given a training dataset $(X, Y)$ consisting of $N$ training points $X = (\boldsymbol{x}_1, \ldots, \boldsymbol{x}_N)$ and the associated labels $Y = (y_1, \ldots, y_N)$, with $\boldsymbol{x}_i \in \mathcal{X} \subseteq \mathbb{R}^n$ and $y_i \in \mathcal{Y} \subseteq \mathbb{R}$.

We focus on NNs defined as the composition of $d$ layers $f = f_d \circ \cdots \circ f_1$, with

$$f_l(\boldsymbol{x}, \boldsymbol{w}) = \rho_l(\boldsymbol{W}_l \, \boldsymbol{x} + \boldsymbol{b}_l) \quad \text{for} \quad l = 1, \ldots, d.$$

Above, $\boldsymbol{W}_l \in \mathbb{R}^{n_l \times n_{l-1}}$ and $\boldsymbol{b}_l \in \mathbb{R}^{n_l}$ with $n_0 = n$ and $n_d = 1$, and $\boldsymbol{w} = (\boldsymbol{W}_1, \boldsymbol{b}_1, \ldots, \boldsymbol{W}_d, \boldsymbol{b}_d)$ are the network's parameters. For all layers but the last, $\rho_l$ will be the ReLU activation function, whereas $\rho_d$ may either be the identity $\rho_d(x) = x$ (regression) or the sigmoid function $\rho_d(x) = 1/(1 + e^{-x})$ (classification).

We optimize $\boldsymbol{w}$ to minimize a differentiable loss function $\ell$ using stochastic gradient descent (SGD). The optimization proceeds in iterations $t$ and each parameter is updated as follows:

$$\boldsymbol{w}^{(t+1)} = \boldsymbol{w}^{(t)} - \alpha_t \frac{\partial \ell(f(\boldsymbol{x}^{(t)}, \boldsymbol{w}^{(t)}), y^{(t)})}{\partial \boldsymbol{w}^{(t)}},$$

where $\boldsymbol{x}^{(t)} \in X$ is a point sampled with replacement from the training set at iteration $t$, $y^{(t)}$ is its label, and $\alpha_t$ is the learning rate. It will also be convenient to refer to $f(\cdot, \boldsymbol{w}^{(t)})$ as $f^{(t)}$.

### 3.1 Linear regions

A well-known property of NNs with ReLU activations is that they partition the input space into regions (convex polyhedra) $\mathcal{R} \subseteq \mathbb{R}^n$ within which $f$ is linear. This viewpoint will be central to our analysis.

There is a simple way to deduce this property from first principles. When $\rho_d$ is the identity, each $f$ can be equivalently expressed as

$$f(\boldsymbol{x}, \boldsymbol{w}) = \boldsymbol{S}_d(\boldsymbol{x})(\boldsymbol{W}_d(\cdots \boldsymbol{S}_2(\boldsymbol{x})(\boldsymbol{W}_2 \boldsymbol{S}_1(\boldsymbol{x})(\boldsymbol{W}_1 \boldsymbol{x} + \boldsymbol{b}_1) + \boldsymbol{b}_2) \cdots) + \boldsymbol{b}_d),$$

where we have defined the input-dependent binary diagonal matrices

$$\boldsymbol{S}_l(\boldsymbol{x}) := \operatorname{diag}(\mathbf{1}[f_l \circ \cdots \circ f_1(\boldsymbol{x}, \boldsymbol{w}) > 0]) \quad \text{and} \quad \boldsymbol{S}_d(\boldsymbol{x}) = 1,$$

with $\mathbf{1}[\boldsymbol{x} > 0]$ being the indicator function applied element-wise. The key observation is that, when the neuron activations $\boldsymbol{S}_l(\boldsymbol{x})$ are fixed for every layer, the above function becomes linear. Thus, each linear region $\mathcal{R}$ of $f$ contains those points that yield the same neuron activation pattern.

Since the activation pattern of any region is uniquely defined by a single point in that region, we write $\mathcal{R}_{\boldsymbol{x}}$ to refer to the region that encloses $\boldsymbol{x}$.

### 3.2 Local and global Lipschitz constants

A function $f$ is Lipschitz continuous with respect to a norm $\| \cdot \|_2$ if there exists a constant $\lambda$ such that for all $\boldsymbol{x}, \boldsymbol{x}'$ we have $\|f(\boldsymbol{x}) - f(\boldsymbol{x}')\|_2 \leq \lambda \|\boldsymbol{x} - \boldsymbol{x}'\|_2$. The minimum $\lambda$ satisfying this condition is called the Lipschitz constant of $f$ and is denoted by $\lambda_f$.

The Lipschitz constant is intimately connected with the gradient. This can be easily seen for differentiable functions $f : \mathcal{X} \to \mathbb{R}$, in which case $\lambda_f = \sup_{\boldsymbol{x} \in \mathcal{X}} \|\boldsymbol{\nabla} f(\boldsymbol{x})\|_2$, where $\mathcal{X}$ is a convex set and $\boldsymbol{\nabla} f(\boldsymbol{x})$ is the gradient of $f$ at $\boldsymbol{x}$ [70, 31, 34].

Although NNs with ReLU activations are not differentiable everywhere, their Lipschitz constant can be determined in terms of their gradient within their regions. Specifically, the local Lipschitz constant within a linear region $\mathcal{R}_{\boldsymbol{x}}$ of $f$ is

$$\lambda_f(\mathcal{R}_{\boldsymbol{x}}) = \|\boldsymbol{\nabla} f(\boldsymbol{x}, \boldsymbol{w})\|_2$$

The Lipschitz constant of $f$ is then simply the largest gradient within any linear region $\lambda_f = \sup_{\boldsymbol{x} \in \mathcal{X}} \|\boldsymbol{\nabla} f(\boldsymbol{x}, \boldsymbol{w})\|_2$. The latter is typically upper bounded by $\lambda_f^{\mathrm{prod}} = \prod_l \|\boldsymbol{W}_l\|_2$ which is known to be a loose bound [21, 22]. For a more formal treatment that also accounts for different types of activation functions and vector-valued outputs, the reader may refer to [22].

## 4 Relating training behavior to NN complexity close to the training data

Our analysis commences in Section 4.1 by deriving a general result that bounds the (appropriately normalized) length of the SGD trajectory over any training interval with the Lipschitz constant of the NN close to training data. Our results on the distance to initialization and weight variance will be implied as corollaries in Section 4.2.

### 4.1 Bounding the length of the SGD trajectory

Theorem 1 formalizes a simple observation: the gradient of a neural network with respect to its input is intimately linked to that with respect to the bias of the first layer. This implies that, by observing how fast the bias of the network is updated, we can deduce what is the Lipschitz constant of the learned function on the linear regions of the input space encountered during training.

**Theorem 1** (Trajectory length). *Let $f^{(t)}$ be a d-layer NN being trained by SGD. Further, denote by*

$$\epsilon_{f^{(t)}}(\boldsymbol{x}, y) := \left| \frac{\partial \ell(\hat{y}, y)}{\partial \hat{y}} \right|_{\hat{y}=f^{(t)}(\boldsymbol{x})}$$

*the gradient of the loss with respect to the NN's output at iteration t. For any set $T$ of iteration indices within which the gradient is not zero, the (normalized) bias trajectory is upper/lower bounded as*

$$\sum_{t \in T} \frac{\lambda_{f^{(t)}}(\mathcal{R}_{\boldsymbol{x}^{(t)}})}{\sigma_1(\boldsymbol{W}_1^{(t)})} \leq \sum_{t \in T} \frac{\|\boldsymbol{b}_1^{(t+1)} - \boldsymbol{b}_1^{(t)}\|_2}{\alpha_t \, \epsilon_{f^{(t)}}(\boldsymbol{x}^{(t)}, y^{(t)})} \leq \sum_{t \in T} \frac{\lambda_{f^{(t)}}(\mathcal{R}_{\boldsymbol{x}^{(t)}})}{\sigma_n(\boldsymbol{W}_1^{(t)})},$$

*where $\sigma_1(\boldsymbol{W}_1^{(t)}) \geq \cdots \geq \sigma_n(\boldsymbol{W}_1^{(t)}) > 0$ are the singular values of $\boldsymbol{W}_1^{(t)}$.*

Theorem 1 shows that lower complexity learners will have a shorter (normalized) bias trajectory. If $\epsilon_{f^{(t)}}(\boldsymbol{x}^{(t)}, y^{(t)})$ and $\alpha_t$ remain approximately constant throughout $T$, the trajectory will grow linearly with the Lipschitz constant of the learner close to the training data.

**Why we focus on the first layer bias.** It might be originally surprising that the gradient w.r.t. $\boldsymbol{b}_1$ is indicative of NN complexity. While the value of $\boldsymbol{b}_1$ is not particularly informative, it turns out that the way it changes over successive SGD iterations reflects the operation of the entire NN: since $\boldsymbol{b}_1$ and $\boldsymbol{x}$ are processed by the NN in a similar fashion, the sensitivity of the NN output w.r.t. changes in the bias relates to those induced by changes in the input. Indeed, via the chain rule, we have

$$\boldsymbol{\nabla} f(\boldsymbol{x}, \boldsymbol{w}) = \boldsymbol{W}_d \boldsymbol{S}_{d-1}(\boldsymbol{x}) \boldsymbol{W}_{d-1} \cdots \boldsymbol{S}_1(\boldsymbol{x}) \boldsymbol{W}_1 = \left( \frac{\partial f(\boldsymbol{x}, \boldsymbol{w})}{\partial \boldsymbol{b}_1} \right)^{\top} \boldsymbol{W}_1,$$

where, for simplicity of exposition, we consider here the case in which $\rho_d$ is the identity function and thus $\boldsymbol{S}_d(\boldsymbol{x}) = 1$. The above equation also explains why the singular values of $\boldsymbol{W}_1$ appear in the bound: since the gradient of $\boldsymbol{b}_1$ does not yield information about $\boldsymbol{W}_1$, we account for it separately. Alternatively, as explained in Appendix C.2, the first layer Lipschitz constant can be controlled by also taking into account the dynamics of $\boldsymbol{W}_1$ and $\boldsymbol{W}_2$. We also note that an identical argument can be utilized to connect the gradient of $\boldsymbol{b}_l$ with the Lipschitz constant of $f_d \circ \cdots \circ f_{l+1}(\boldsymbol{x})$.

**Understanding the normalization.** The normalization by $\alpha_t \, \epsilon_{f^{(t)}}(\boldsymbol{x}^{(t)}, y^{(t)})$ renders the bound independent of the learning rate $\alpha_t$ as well as of how well the network fits the training data. When a mean-squared error (MSE) and a binary cross-entropy (BCE) loss is employed

$$\ell_{\text{MSE}}(\hat{y}, y) = \frac{(\hat{y} - y)^2}{2} \quad \text{and} \quad \ell_{\text{BCE}}(\hat{y}, y) = -y \log(\hat{y}) - (1 - y) \log(1 - \hat{y}),$$

with $y, \hat{y} \in \mathbb{R}$ and $y, \hat{y} \in [0, 1]$, respectively, we have

$$\epsilon_f(\boldsymbol{x}, y) = |f(\boldsymbol{x}) - y| \quad \text{and} \quad \epsilon_f(\boldsymbol{x}, y) = \frac{1}{|1 - y - f(\boldsymbol{x})|}.$$

In both cases, $\epsilon_f(\boldsymbol{x}, y)$ measures the distance between the true label and the NN's output.

**Applicability to other architectures.** Beyond fully-connected layers, Theorem 1 directly applies to layers that involve weight sharing and/or sparsity constraints, such as convolutional and locally-connected layers, as long as $\boldsymbol{b}_1$ remains non-shared. In addition, the result also holds unaltered for networks that utilize skip connections or max/average pooling after the 1st layer, as well as for NNs with general element-wise activation functions (see Appendix C.1).

**Dependence on the singular values of $\boldsymbol{W}_1$.** The lower bound presented in Theorem 1 can be expected to be much tighter than the upper bound since the largest singular value is usually a reasonably small constant, whereas the smallest may be (close to) zero. Fortunately, the upper bound can be tightened when the data fall within some lower-dimensional space $\mathcal{S}$. In that case, one may substitute the minimum singular value with the minimum of $\|\boldsymbol{W}_1^{(t)} \boldsymbol{x}\|_2$ for all $\boldsymbol{x} \in \mathcal{S}$ of unit norm.

### 4.2 Corollaries: steady learners, variance of bias, and distance to initialization

Suppose that after some iteration our NN has fit the training data relatively well. We will say that the NN is a "steady learner" if its 1st layer bias is updated slowly:

**Definition 1** (Steady learner). *A NN $f^{(t)}$ trained by SGD is $(\tau, \varphi)$-steady if*

$$\frac{\|\boldsymbol{b}_1^{(t+1)} - \boldsymbol{b}_1^{(t)}\|_2}{\alpha_t \, \epsilon_{f^{(t)}}(\boldsymbol{x}^{(t)}, y^{(t)})} \leq \varphi \quad \textit{for all} \quad t \geq \tau.$$

The following is a simple corollary:

**Corollary 1.** *Let $f^{(t)}$ be $(\tau, \varphi)$-steady. Consider an interval $T$ of iterations after $\tau$ and suppose that $\sigma_1(\boldsymbol{W}_1^{(t)}) \leq \beta$ for every $t \in T$. Select an iteration $t \in T$ at random. The Lipschitz constant of $f^{(t)}$ at every training point $\boldsymbol{x} \in X$ will be bounded by $\lambda_{f^{(t)}}(\mathcal{R}_{\boldsymbol{x}}) \leq \beta \varphi$, generically, i.e., with probability that converges to 1 as $|T|$ grows.*

Crucially, the bound of Corollary 1 can be exponentially tighter than the product-of-norms bound $\lambda_f^{\text{prod}}$: whereas $\beta\varphi$ does not generally depend on depth, $\lambda_f^{\text{prod}}(\mathcal{R}_{\boldsymbol{x}}) = w^d$ when $\|\boldsymbol{W}_l\|_2 = w$.

We will also use Theorem 1 to characterize two other aspects of the training behavior: the parameter variance as well as the distance to initialization. The following corollary shows that the weights of high complexity NNs cannot concentrate close to some local minimum:

**Corollary 2** (Variance of bias). *Let $f^{(t)}$ be a $d$-layer NN with ReLU activations being trained by SGD. Let $T$ be a set of iteration indices and write*

$$\epsilon_{harm}(T) := \sqrt{\operatorname*{harm}_{t \in T} \epsilon_{f^{(t)}}(\boldsymbol{x}^{(t)}, y^{(t)})^2}$$

*for the square-root of the harmonic mean of the squared loss derivatives within $T$. Then, the bias of the first layer will exhibit variance at least:*

$$\operatorname*{avg}_{t \in T} \|\boldsymbol{b}_1^{(t)} - \operatorname*{avg}_{t \in T} \boldsymbol{b}_1^{(t)}\|_2^2 \geq \left( \operatorname*{avg}_{t \in T} \frac{\alpha_t \, \lambda_{f^{(t)}}(\mathcal{R}_{\boldsymbol{x}^{(t)}}) \, \epsilon_{harm}(T)}{2 \, \sigma_1(\boldsymbol{W}_1^{(t)})} \right)^2. \tag{1}$$

Therefore, a larger complexity NN will need to fit the training data more closely (so that $\epsilon_{\text{harm}}(T)$ decreases) to achieve the same variance as that of a lower complexity NN.

We can also deduce that the bias will remain closer to initialization for NNs that have a smaller Lipschitz constant:

**Corollary 3** (Distance to initialization). *Let $f^{(t)}$ be a $d$-layer NN being trained by SGD with an MSE loss and fix some iteration $\tau$. The first layer bias may move from its initialization by at most*

$$\|\boldsymbol{b}_1^{(\tau)} - \boldsymbol{b}_1^{(0)}\|_2 \leq \sum_{t=0}^{\tau-1} \alpha_t \, \frac{\epsilon_{f^{(t)}}(\boldsymbol{x}^{(t)}, y^{(t)}) \, \lambda_{f^{(t)}}(\mathcal{R}_{\boldsymbol{x}^{(t)}})}{\sigma_n(\boldsymbol{W}_1^{(t)})}.$$

The latter result is only meaningful in a regression setting. When using the BCE loss, the loss derivative can grow exponentially when the classifier is confidently wrong. On the contrary, with an MSE loss in place the loss derivative grows only linearly with the error, rendering the bound more meaningful.

When $\alpha_t$ and $\epsilon_{f^{(t)}}(\boldsymbol{x}^{(t)}, y^{(t)})$ decay sufficiently fast, the bound depends on $\sigma_n(\boldsymbol{W}_1^{(t)})$ and the (normalized) Lipschitz constant at and close to initialization. Therefore, the corollary asserts that SGD with an MSE loss can find solutions near to initialization if two things happen: the NN fits the data from relatively early on in the training while implementing a low-complexity function close to the training data.

## 5 NN complexity far from the training data

Our exploration on the relationship between training and the complexity of the learned function thus far focused only on regions of the input space that contain at least one training point. It is natural to ask how the function behaves in empty regions. After all, to make generalization statements we need to ensure that the learned function has, with high probability, bounded Lipschitz constant close to any point in the training distribution.

Next, we provide conditions such that a NN that undergoes steady bias updates, as per Definition 1, also has low complexity in linear regions that do not contain any training points. Our analysis starts in Section 5.1 by relating the Lipschitz constant of regions in and outside the training data. We then show in Section 5.2 how learners that remain steady while trained with Dropout have a bounded generalization error.

A central quantity in our analysis is the activation $s_t(\boldsymbol{x})$ associated with each $\boldsymbol{x}$:

$$\boldsymbol{s}_t(\boldsymbol{x}) := \bigotimes_{l=d-1}^{1} \text{diag}\left(\boldsymbol{S}_l^{(t)}(\boldsymbol{x})\right) = \bigotimes_{l=d-1}^{1} \mathbf{1}[f_l \circ \cdots \circ f_1(\boldsymbol{x}, \boldsymbol{w}) > 0] \in \{0, 1\}^{n_{d-1} \cdots n_1}$$

Thus, $\boldsymbol{s}_t(\boldsymbol{x})$ is the Kronecker product of all activations when the NN's input is $\boldsymbol{x}$.

We will also assume that the learned function $f^{(t)}$ eventually becomes consistent on the training data:

**Assumption 1.** *There exists $\tau, \gamma > 0$ such that $\boldsymbol{s}_t(\boldsymbol{x}) = \boldsymbol{s}_{t'}(\boldsymbol{x})$ and $\lambda_{f^{(t)}}(\mathcal{R}_{\boldsymbol{x}}) \leq (1 + \gamma) \lambda_{f^{(t')}}(\mathcal{R}_{\boldsymbol{x}})$ for all $\boldsymbol{x} \in X$ and $t, t' \geq \tau$.*

Assumption 1 is weaker than requiring that the parameters have converged: the parameters are allowed to keep changing as long as the slope and activation pattern on each training point remains similar. We also stress that the NN can still have different activation patterns at different points (and thus be highly non-linear), as long as these activations stay persistent over successive iterations. Naturally, it is always possible to satisfy our assumption by decaying the learning rate appropriately.

## 5.1 The Lipschitz constant of empty regions

As we show next, the Lipschitz constant of a NN can be controlled for those regions whose neural activation can be written as a combination of activations of training points.

**Theorem 2.** *Let $T$ be any interval of SGD iterations that satisfies Assumption 1, and suppose that $\sigma_1(\boldsymbol{W}_1^{(t)}) \leq \beta$ for all $t \in T$. Furthermore, denote, respectively, by*

$$\boldsymbol{S}_T := \left[\boldsymbol{s}_t(\boldsymbol{x}^{(t)})\right]_{t \in T}, \quad \boldsymbol{\varphi}_T := \left[\frac{\|\boldsymbol{b}_1^{(t+1)} - \boldsymbol{b}_1^{(t)}\|_2}{\alpha_t \, \epsilon_{f^{(t)}}(\boldsymbol{x}^{(t)}, y^{(t)})}\right]_{t \in T}, \quad \mu_T := \min_{t \in T}\{f^{(t)}(\boldsymbol{x}^{(t)}), 1 - f^{(t)}(\boldsymbol{x}^{(t)})\}$$

*the binary matrix whose columns are the neural activations of all points sampled within $T$, the vector containing the normalized bias updates, and the distance to integrality if a sigmoid is used in the last layer. Select a point $\boldsymbol{x} \in \mathbb{R}^n$ that is not in the training set. For all $t \in T$, the Lipschitz constant of $f^{(t)}$ in $\mathcal{R}_{\boldsymbol{x}}$ is bounded by the following Basis Pursuit problem:*

$$\lambda_{f^{(t)}}(\mathcal{R}_{\boldsymbol{x}}) \leq (1 + \gamma) \beta \xi \min_{\boldsymbol{a}} \|\boldsymbol{a} \odot \boldsymbol{\varphi}_T\|_1 \quad \text{subject to} \quad \boldsymbol{s}_t(\boldsymbol{x}) = \boldsymbol{S}_T \, \boldsymbol{a},$$

*where $\odot$ is the Hadamard product, $\xi = \frac{0.25}{\mu_T(1 - \mu_T)}$ if a sigmoid is used and $\xi = 1$, otherwise.*

To grasp an intuition of the bound, suppose that we are in a regression setting ($\xi = 1$) and that the interval $T$ is large enough so that we have seen all training points. Theorem 2 then implies:

$$\exists \boldsymbol{x}_{i_1}, \ldots, \boldsymbol{x}_{i_k} \in X, \, \boldsymbol{s}_t(\boldsymbol{x}) = \boldsymbol{s}_t(\boldsymbol{x}_{i_1}) + \cdots + \boldsymbol{s}_t(\boldsymbol{x}_{i_k}) \implies \lambda_f^{(t)}(\mathcal{R}_{\boldsymbol{x}}) \leq k \, \beta(1 + \gamma) \, \|\boldsymbol{\varphi}_T\|_\infty.$$

By itself, Theorem 2 does not suffice to ensure that the function is globally Lipschitz because the theorem does not have predictive power for points $\boldsymbol{x}$ whose activation $\boldsymbol{s}_t(\boldsymbol{x})$ cannot be written as a linear combination $\boldsymbol{S}_T \boldsymbol{a}$ of activations of the training points. A sufficient condition for the theorem to yield a global bound is that $\boldsymbol{S}_T$ is full rank, but the latter can only occur for $N \geq n_d \cdots n_1$, a quantity that grows exponentially with the depth of the network. Section 5.2 will provide a significantly milder condition for networks trained with Dropout.

## 5.2 Learners that remain steady with Dropout generalize

Dropout entails deactivating each active neuron independently with probability $p$. We here consider the variant that randomly deactivates each neuron independently[2] at the end of the forward-pass with

---

[2]Typically, the neurons are dropped during the forward pass and not at the end as we do here. However, for networks with $d \leq 2$, the two mechanisms are identical.

probability $1/2$. We focus on binary NN classifiers

$$g^{(t)}(\boldsymbol{x}) := \mathbf{1}\left[f^{(t)}(\boldsymbol{x}) > 0.5\right]$$

trained with a BCE loss, and with the NN's last layer using a sigmoid activation. The empirical and expected classification error is, respectively, given by

$$\text{er}_t^{\text{emp}} = \underset{i=1}{\overset{N}{\text{avg}}}\, \mathbf{1}\left[g^{(t)}(\boldsymbol{x}_i) \neq y_i\right] \quad \text{and} \quad \text{er}_t^{\text{exp}} = \text{E}_{(\boldsymbol{x},y)}\left[\mathbf{1}\left[g^{(t)}(\boldsymbol{x}) \neq y\right]\right].$$

Theorem 3 controls the generalization error in terms of the number $\mathcal{N}(\mathcal{X}; \ell_2, r_t(X))$ of $\ell_2$ balls of radius $r_t(X)$ needed to cover the data manifold $\mathcal{X}$. The radius is shown to be larger for more steadily trained classifiers (through $1/\varphi$) and to depend logarithmically on the number of neurons:

**Theorem 3.** *Let $f^{(t)}$ be a depth $d$ NN with ReLU activations being trained with SGD, a BCE loss and $1/2$-Dropout.*

*Suppose that $f^{(t)}$ is $(\tau, \varphi)$-steady and that for every $t \geq \tau$ the following hold: (a) Assumption 1, (b) $\boldsymbol{s}_t(\boldsymbol{x}) \leq \sum_{i=1}^{N} \boldsymbol{s}_t(\boldsymbol{x}_i)$ for every $\boldsymbol{x} \in \mathcal{X}$, (c) $\sigma_1(\boldsymbol{W}_1^{(t)}) \leq \beta$, and (d) $f^{(t)}(\boldsymbol{x}^{(t)}) \in [\mu, 1 - \mu]$. Define*

$$r_t(X) = \frac{\min_{i=1}^{N}|1 - 2f^{(t)}(\boldsymbol{x}_i)|}{c\,\varphi\,\log\left(\sum_{l=1}^{d-1} n_l\right)} \quad \text{and} \quad c = \frac{(1+\gamma)\,\beta\,(1+o(1))}{\mu\,(1-\mu)\,p_{min}},$$

*where $p_{min} = \min_{l<d, i \leq n_l, t \geq \tau}[\text{avg}_{\boldsymbol{x} \in X}\, diag(\boldsymbol{S}_l^{(t)}(\boldsymbol{x}))]_i > 0$ is the minimum frequency that any neuron is active before Dropout is applied.*

*For any $\delta > 0$, with probability at least $1 - \delta$ over the Dropout and the training set sampling, the generalization error is at most*

$$|er_t^{emp} - er_t^{exp}| = O\left(\sqrt{\frac{\mathcal{N}(\mathcal{X}; \ell_2, r_t(X)) + \log(1/\delta)}{N}}\right),$$

*where $\mathcal{N}(\mathcal{X}; \ell_2, r)$ is the minimal number of $\ell_2$-balls of radius $r$ needed to cover $\mathcal{X}$.*

**Intuition.** Recall that the bounds derived in Section 4 only concern the regions of the NN that contain at least one training point. However, due to the geometry of these regions, it is possible (and likely) that small empty regions will be located near those that have training data inside them. Deriving a generalization bound would require upper bounding the Lipschitz constant of the NN on these empty regions. Unfortunately, to our knowledge, the latter is not possible without resorting to loose product-of-norms bounds or imposing strong additional assumptions that assert that the training regions are sufficiently diverse (as Theorem 2 does). Here is where Dropout comes in. Dropout introduces stochasticity in the training procedure that provides information of the NN function in the gaps between training data. This allows us to infer the Lipschitz constant of the function in larger portions of the space from the training behavior — and thus to relax the assumptions of Theorem 2. Specifically, the proof approximates the global Lipschitz constant as follows:

$$\lambda_{f^{(t)}}^{\text{steady}} := \frac{c\,\varphi}{4}\,\log\left(\sum_{l=1}^{d-1} n_l\right) \quad \text{with} \quad \lambda_{f^{(t)}} \leq \lambda_{f^{(t)}}^{\text{steady}} \leq \lambda_{f^{(t)}} \cdot O\left(\log\left(\sum_{l=1}^{d-1} n_l\right)\right).$$

It then invokes a robustness argument [18, 19] to control the generalization error. The bound above comes in sharp contrast with the product-of-norms bound $\lambda_f^{\text{prod}}$, which grows exponentially with $d$ and can be arbitrary larger than $\lambda_{f^{(t)}}$, since there exists parameters for which $\lambda_f = 0$ and $\lambda_f^{\text{prod}} > 0$.

**Understanding the assumptions made.** The strongest requirement posed by Theorem 3 is that every neural pathway is activated for each training point: $\boldsymbol{s}_t(\boldsymbol{x}) \leq \sum_{i=1}^{N} \boldsymbol{s}_t(\boldsymbol{x}_i)$ for every $\boldsymbol{x} \in \mathcal{X}$. In contrast to Theorem 2, the latter can be satisfied even when $N = 1$, e.g., if there exist some training point for which all neurons are active. However, the assumption will not hold when some entries of $\boldsymbol{s}_t(\boldsymbol{x})$ are never activated after iteration $\tau$. Little can also be said about the global behavior of $f^{(t)}$ when there are neurons that are not periodically active (which would also imply $p_{min} = 0$). We

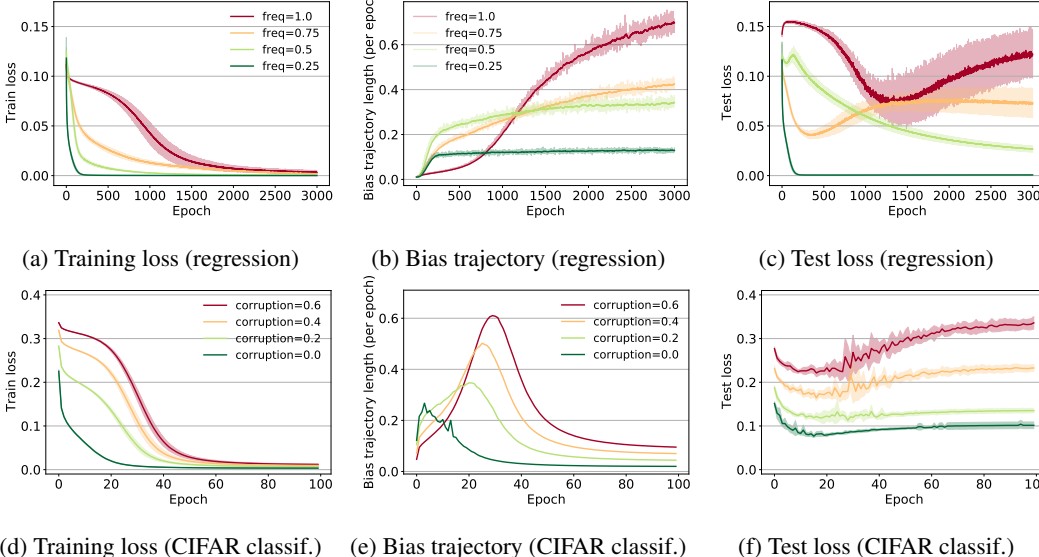

(a) Training loss (regression)  (b) Bias trajectory (regression)  (c) Test loss (regression)

(d) Training loss (CIFAR classif.)  (e) Bias trajectory (CIFAR classif.)  (f) Test loss (CIFAR classif.)

Figure 2: Training behavior of MLP (top) and CNN (bottom) solving a task of increasing complexity (green→red): fitting a function of increasing spatial frequency (top) and classifying CIFAR images with increasing label corruption (bottom). In accordance with Theorem 1, the per epoch bias trajectory (middle subfigures) is longer when the network is asked to fit a more complex training set.

argue however that such neurons can be eliminated without any harm as, by definition, they are not affecting the NN's output after $\tau$.

**Dependence on the classifier's confidence.** According to Theorem 3, the best generalization is attained when the classifier has some certainty about its decisions on the training set (so that $|1 - 2f^{(t)}(\boldsymbol{x}_i)| = \Omega(1)$), while also not being overconfident (so that $\mu(1 - \mu) = O(1)$).

**Dependence on the data distribution and the number of parameters.** A interesting property of the bound is that it depends on the intrinsic dimension of the data rather than the ambient dimension. For instance, if $\mathcal{X}$ is a $C_M$-regular $k$-dimensional manifold with $C_M = O(1)$ it is known [71] that

$$\mathcal{N}(\mathcal{X}; \ell_2, r) = \left(\frac{C_M}{r}\right)^k, \quad \text{implying that} \quad N = O(r_t(X)^{-k})$$

training points suffice to ensure generalization. This sample complexity bound grows poly-logarithmically with the number of neurons $n_d \cdots n_1$ and the number of parameters when $c\varphi = O(1)$. On the contrary, since the radius $r$ of the $\ell_2$ balls used in the covering grows inversely proportionally to the Lipschitz constant, if the product-of-norms bound was used in our proof, then the sample complexity would be exponentially larger: if $\|\boldsymbol{W}_l\|_2 = w$ then $r = O(w^{-d})$ and $N = O(w^{dk})$.

As remarked by Sokolić et al. [19], other data distributions with covering numbers that grow polynomially with $k$ include rank-$k$ Gaussian mixture models [72] and $k$-sparse signals under a dictionary [73].

## 6 Experiments

We test our findings in the context of two tasks:

**Task 1. Regression of a sinusoidal function with increasing frequency.** In this toy problem, a multi-layer perceptron (MLP) is tasked with fitting a randomly-sampled 2D sinusoidal function with increasing frequency (0.25, 0.5, 0.75, 1) isometrically embedded in a 10-dimensional space. More details can be found in Appendix A.1. The setup allows us to test our results while precisely controlling the complexity of the ground-truth function: fitting a low-frequency function necessitates a smaller Lipschitz constant than a high-frequency one. We trained an MLP with 5 layers consisting entirely of ReLU activations and with the 1st layer weights being identity. We repeated the experiment

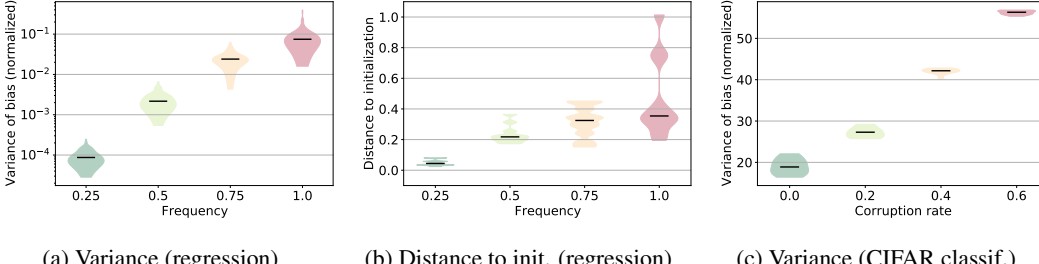

(a) Variance (regression)  (b) Distance to init. (regression)  (c) Variance (CIFAR classif.)

Figure 3: A closer inspection of how the bias is updated. The variance is computed over the last 10 epochs. As seen, the bias of higher complexity NNs varies more close to convergence (Corollary 2). Further, with an MSE loss, high complexity NNs may veer off further from initialization (Corollary 3).

10 times, each time training the network with SGD using a learning rate of 0.001 and an MSE loss until it had fit the sinusoidal function at 100 randomly generated training points.

**Task 2. CIFAR classification under label corruption.** In our second experiment, we trained a convolutional neural network (CNN) to classify 10000 images from the 'dog' and 'airplane' classes of CIFAR10 [74]. The classes were selected at random. We focus on binary classification to remain consistent with the theory. Inspired by [4], we artificially increase the task complexity by randomly corrupting a (0, 0.2, 0.4, 0.6) fraction of the training labels. Thus, a higher corruption implies a larger complexity function. Differently from the first task, we used a CNN with 2 convolutional layers featuring ReLU activations in intermediate layers and a sigmoid activation in the last. We set the first layer identically with the regression experiment. We repeated the experiment 8 times, each time training the network with SGD using a BCE loss and a learning rate of 0.0025.

In agreement with previous studies [4, 6, 3, 2], Figure 2 shows that training slows down as the complexity of the fitted function increases. Figures 2b and 2e depict the per-epoch bias trajectory: $\sum_{t \in T_{\text{epoch}}} \|\boldsymbol{b}_1^{(t+1)} - \boldsymbol{b}_1^{(t)}\|_2 / \alpha_t \epsilon_{f^{(t)}}(\boldsymbol{x}^{(t)}, y^{(t)})$ According to Theorem 1, this measure captures the Lipschitz constant $\lambda_{f^{(t)}}(\mathcal{R}_{\boldsymbol{x}^{(t)}})$ of the NN during each epoch and across all training points. In agreement with our theory, the bias trajectory is significantly longer when fitting higher complexity functions. The length of the total trajectory is the integral of the depicted curve, see Appendix A.4. Moreover, as shown in Fig 2c, the trajectory length also correlates with the loss of the network on a held-out test set, with longer trajectories consistently corresponding to poorer test performance.

We proceed to examine more closely the behavior of $\boldsymbol{b}_1^{(t)}$ during training. Figures 3a and 3b corroborate the claims of Corollaries 2 and 3, respectively: when fitting a lower complexity function and an MSE loss is utilized, the bias will remain more stable (here we show the variance in the last 10 epochs) and closer to initialization. The same variance trend can be seen in Figure 3c for image classification. The distance-to-initialization analysis is not applicable to classification (due to the BCE gradient being unbounded), but we include the figure in Appendix A.2 for completeness.

**Additional results.** The interested reader can refer to Appendices A.3 and A.4 for visualizations of the Lipschitz constants within linear regions and of the total bias trajectory length. Appendices A.6 and A.5 test how our findings are affected by the batch size and architecture, whereas Appendix A.7 examines the training dynamics associated with deeper layer biases.

# 7 Conclusion

This paper showed that the training behavior and the complexity of a NN are interlinked: networks that fit the training set with a small Lipschitz constant will exhibit a shorter bias trajectory and their bias will vary less. Though our study is of primarily theoretical interest, our results provide support for the Benevolent Training Hypothesis and suggest that favoring NNs that exhibit good training behavior can be a useful bias towards models that generalize well.

At the same time, there are many aspects of the BHT that we do not yet understand: what is the effect of optimization algorithms and of batching on the connection between complexity and training behavior? Does layer normalization play a role? What can be glimpsed by the trajectory of other parameters? We believe that a firm understanding of these questions will be essential in fleshing out the interplay between training, NN complexity, and generalization.

## Acknowledgments and Disclosure of Funding

We are thankful to the anonymous reviewers, Giorgos Bouritsas, Martin Jaggi, Nikos Karalias, Mattia Atzeni, and Jean-Baptiste Cordonnier for engaging in fruitful discussions and providing valuable feedback. Andreas Loukas would like to thank the Swiss National Science Foundation for supporting him in the context of the project "Deep Learning for Graph Structured Data", grant number PZ00P2 179981. Stefanie Jegelka acknowledges funding from NSF CAREER award 1553284, NSF BIGDATA award 1741341 and an MSR Trustworthy and Robust AI Collaboration award.

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
