# Supplementary material: what training reveals about neural network complexity

**Andreas Loukas**
EPFL
andreas.loukas@epfl.ch

**Marinos Poiitis**
Aristotle University of Thessaloniki
mpoiitis@csd.auth.gr

**Stefanie Jegelka**
MIT
stefje@mit.edu

The supplementary material commences in Section B by presenting the proofs supporting our theoretical claims. Section A discusses experimental details and shows additional results. Finally, Section C lays out theoretical results of side-interest.

## A    Additional empirical results

### A.1    Description of Task 1

The input data of Task 1 are generated by the following two step procedure:

First, we sample $N = 100$ points $z_i \in [-1, 1]^2$ uniformly at random and assign them a ground truth label according to the sinusoidal function:

$$y_i = \cos(2\pi\omega\, z_i(1)) \cdot \cos(2\pi\omega\, z_i(1)) \in [-1, 1],$$

where $\omega$ is interpreted as a frequency and we set $\omega \in \{0.25, 0.5, 0.75, 1.0\}$ in our experiments. The four resulting functions are visualized in Figure A.2.

We then determine $\{x_i\}_{i=1}^N$ by isometrically embedding $\{z_i\}_{i=1}^N$ into $\mathbb{R}^{10}$. We achieve this by selecting the first 2 columns $R \in \mathbb{R}^{10 \times 2}$ of a random $10 \times 10$ unitary matrix and setting $x_i = R\, z_i$. This procedure ensures that the distances between points remains the same in high dimensions.

### A.2    Distance to initialization for Task 2

We focus on the image classification CNN trained with a BCE loss. Figure A.1 depicts the distance from initialization $\|b_1^{(t)} - b_1^{(0)}\|_2$ in the last 10 training epochs.

As explained in Section 4.2, when a BCE loss is utilised, the derivative of the loss becomes unbounded which stops Corollary 3 from applying. Interestingly, Figure A.1 confirms this by showing that the distance is not an increasing function of complexity. The reverse phenomenon can be observed when an MSE loss is utilized (see Figure 3b).

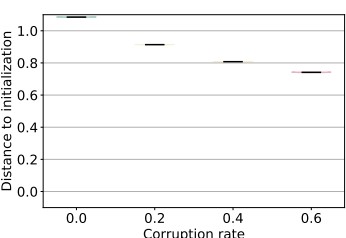

Figure A.1: Distance to init. with BCE loss.

### A.3    Visualizing linear regions

Aiming to gain intuition about the behavior of NNs in linear regions close and far the training data, we take a closer look at the function an MLP is learning when trained to solve task 1 ($\omega = 0.5$, 2 hidden layers, $N = 200$).

Figures A.3a and A.3b depict, respectively, the real and learned function projected in 2D (recall that the true function is isometrically embedded in 10D). Blue dots are training data points. The

35th Conference on Neural Information Processing Systems (NeurIPS 2021).

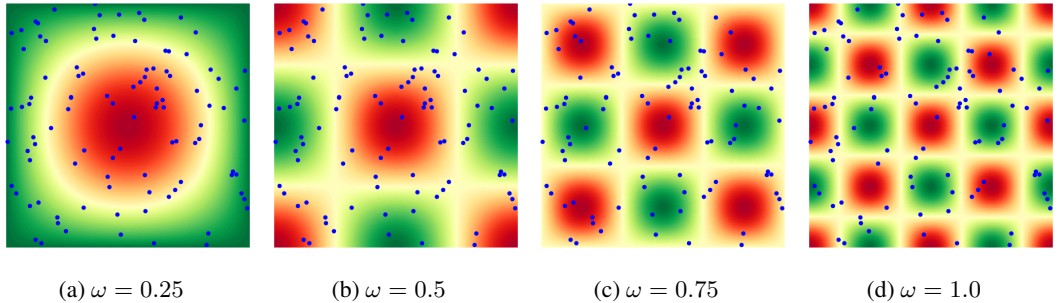

(a) $\omega = 0.25$   (b) $\omega = 0.5$   (c) $\omega = 0.75$   (d) $\omega = 1.0$

Figure A.2: The surface of the sinusoidal function from where the input points are sampled, for different frequencies $\omega$. Sampled points are plotted on top of the surface.

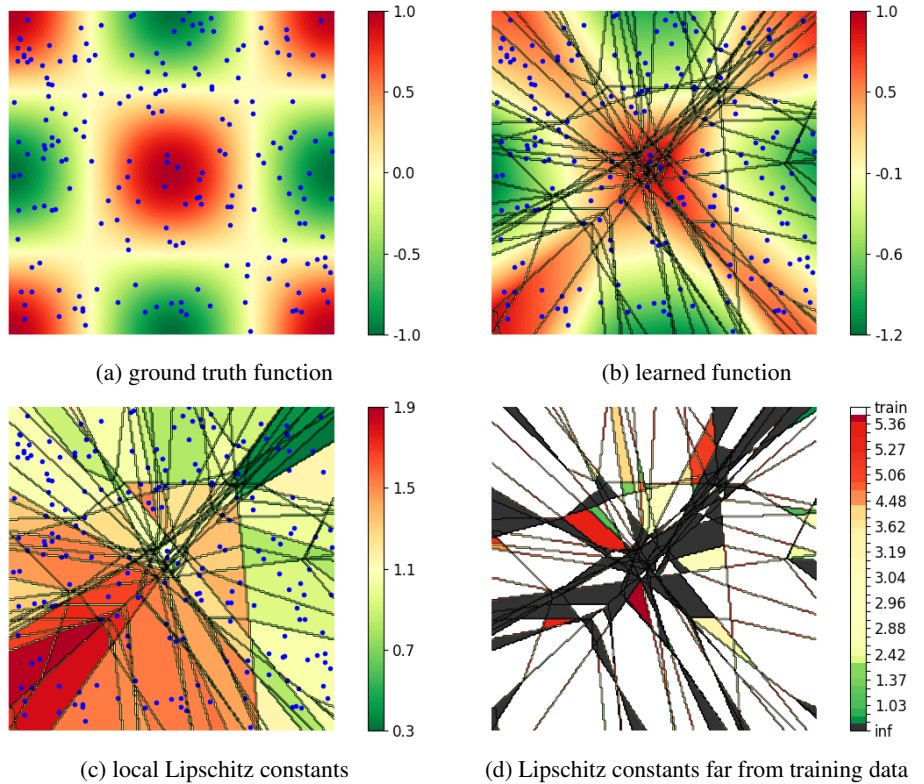

(a) ground truth function   (b) learned function

(c) local Lipschitz constants   (d) Lipschitz constants far from training data

Figure A.3: Visual illustration of the linear regions (top right) of the trained NN when fitting a sinusoidal function (top left), together with their local Lipschitz constants (bottom left) and those far from the training data as predicted by Theorem 2 (bottom right).

boundaries between region are indicated with black lines. As observed, there is a large number of linear regions of varying sizes with the smaller and more densely packed regions being found close to the (0,0) point.

The bottom two panels display the local Lipschitz constants (i.e., the magnitude of the gradient within every region). In Figure A.3c we can see the real constants at all regions. Interestingly, it appears that low- and high-Lipschitz constants are clustered, which likely follows from the hierarchical region formation process: in other words, regions within the same cluster fall within the same region of a shallower sub-network and are split by a higher layer.

Figure A.3d distinguishes between regions containing training points (in white) and the rest (in color). We color empty regions depending on the bound given by Theorem 2 and black regions are those for which the theorem does not have predictive power. We observe that, though the proposed theory

allows us to make statements about the function behavior far from the training data, the theory does not explain the global behavior of the NN. This motivates the introduction of Dropout in the analysis of Section 5.2: by exploiting stochasticity we can infer more properties about the NN complexity from the training trajectory. Intuitively, using Dropout during a sufficiently long training, one is able to deduce from the observed bias updates (specifically vector $\varphi_T$ in Theorem 2) the Lipschitz constants within more regions (thus they would also bound the Lipschitz constant within some of the non-white regions in Fig A.3d). Moreover, though encountering each and every region in the training would likely take a very long time, Theorem 2 implies that only a small subset of regions suffice to approximate the global Lipschitz constant up to a logarithmic factor.

We finally observe that, in the regions were it applies, Theorem 2 yields a bound that is a constant factor away from the real local Lipschitz constants: the bound overestimates the constants by roughly a factor of four.

## A.4   Total trajectory length

Figure A.4 displays the length of the entire normalized bias trajectory at every point in the training. Thus, Figure A.4 corresponds to the integral of Figures 2b and 2e, which focus on the length of the normalized bias trajectory within every epoch. We note also that all NNs have been trained until they could closely fit the training set.

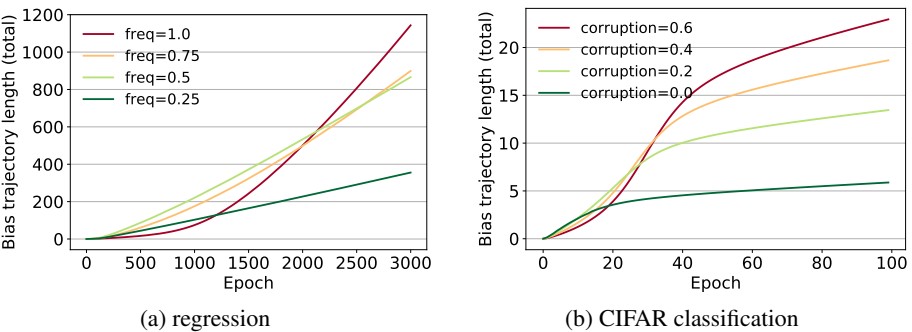

(a) regression                      (b) CIFAR classification

Figure A.4: Length of the normalized trajectory at every pointing in the training starting from initialization. As expected, the trajectory length of NNs grows with the complexity of the function they are learning.

A side-by-side comparison with Figures 2a and 2d reveals that, between any two NNs that have fitted the training data equally well, the one that implements a higher complexity function has consistently a longer trajectory.

## A.5   Effect of architecture on bias trajectory

We next evaluate the effect of the NN architecture on the optimization trajectory. We focus on the MNIST dataset [1] and train an MLP and a CNN to distinguish between digits '3' and '6' based on a training set consisting of 100 and 1000 images per class. For consistency with the previous experiments, we used the same NN architectures for the MLP and CNN as those employed for Tasks 1 and 2, respectively (though both NNs now feature a sigmoid activation in the last layer). The networks are trained using SGD with a BCE loss and a learning rate of $\alpha_t = 0.002$.

Figure A.5 depicts the training loss, normalized bias trajectory length, and test loss for each dataset. Note that, in contrast to Figure 2, here all three measures are computed over time-intervals of 100 iterations (rather than per epoch). As expected, when the training set is small, both architectures fit the training data equally well after roughly 20k iterations, but the CNN overfits less. By observing the length of the bias trajectory, we deduce that the MLP is learning a more complex function than the CNN. Thus, in the MNIST100 case, there is a correlation between trajectory length and generalization with the NN architecture that is more appropriate for the task exhibiting a shorter trajectory.

It is important to remark that the complexity of the learned function is not the sole factor driving generalization (though it is can be a crucial factor all other things being equal). Convolutional layers

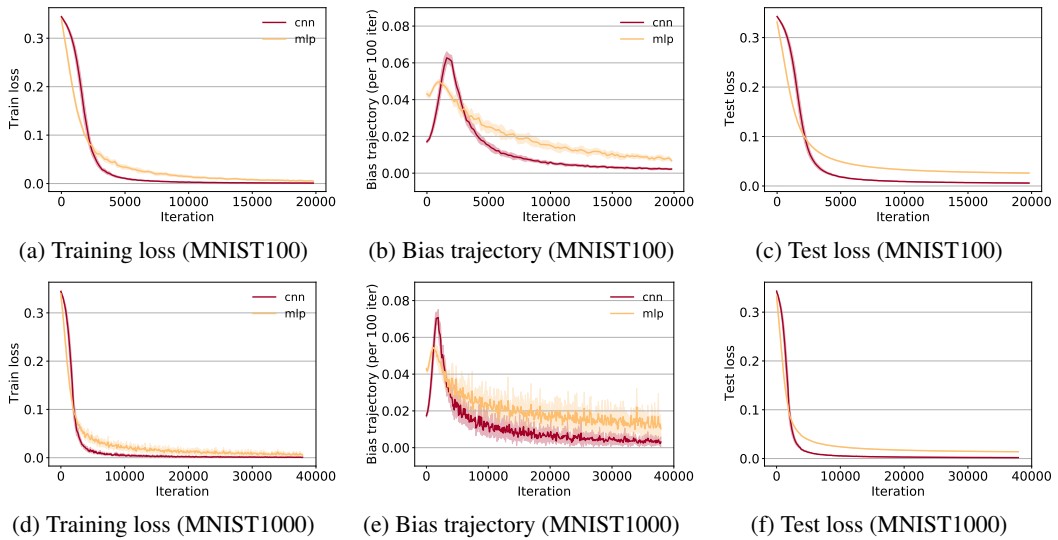

(a) Training loss (MNIST100)  (b) Bias trajectory (MNIST100)  (c) Test loss (MNIST100)

(d) Training loss (MNIST1000)  (e) Bias trajectory (MNIST1000)  (f) Test loss (MNIST1000)

Figure A.5: Training and test behavior of a MLP and a CNN solving a binary MNIST image classification task with different number of samples (100 and 1000 images per class). All three measures (training loss, bias trajectory length, and test loss) are plotted at intervals of 100 iterations.

are indeed more constrained and better suited to image data than fully convolutional ones – thus it is reasonable to expect better generalization than MLPs. Nevertheless, our experiment shows that the CNN, beyond having the right architecture for the task, also learns a slightly lower complexity function than the MLP while fitting the training data equally well or better. Thus, here we mainly use the bias trajectory length as a diagnostic tool that helps us understand what functions the two architectures are learning.

## A.6 Effect of batch size on bias trajectory

This experiment investigates the effect of different batch sizes on the bias trajectory. We adopt the same setup as that of Task 1 (specifically $\omega = 0.25$ and $0.75$ in Figure A.6) and train NNs with SGD using batch sizes of 16 and 32, whereas our original experiment used a batch size of 1.

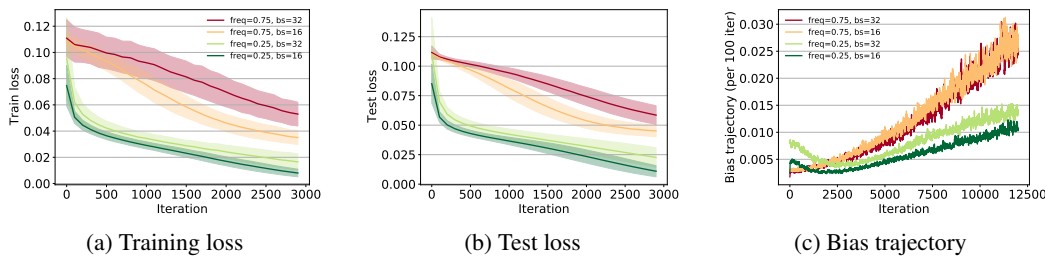

(a) Training loss  (b) Test loss  (c) Bias trajectory

Figure A.6: Training and test behavior of MLPs using different batch size (16 and 32) and evaluated on data sampled from two different frequencies ($\omega = 0.25$ and $0.75$ in Task 1). All three measures (training loss, bias trajectory length, and test loss) are plotted at intervals of 100 iterations. The trajectory length correlates with the NN's complexity for different batch sizes.

The results are consistent with those of Figure 2, with a longer bias trajectory indicating that the NN is fitting a more complex hypothesis and correlating with higher test loss. Increasing the batch size from 1 to 32 also leads to a slight increase in trajectory length, though we currently lack mathematical evidence that support this empirical observation.

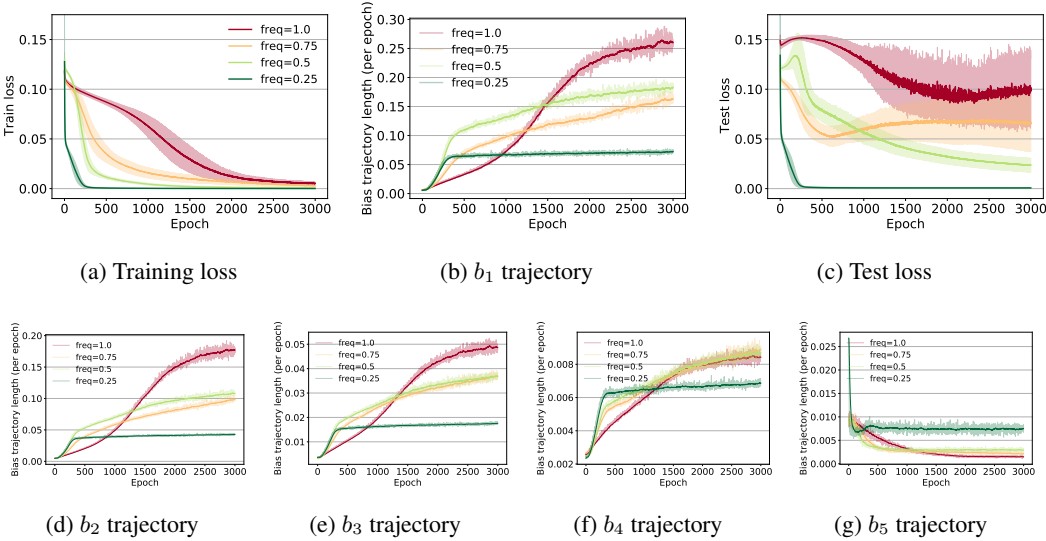

(a) Training loss        (b) $b_1$ trajectory        (c) Test loss

(d) $b_2$ trajectory    (e) $b_3$ trajectory    (f) $b_4$ trajectory    (g) $b_5$ trajectory

Figure A.7: Illustration of how the biases of all layers change when training a MLP to fit a function of increasing spatial frequency. Sub-figures (a) and (c) show the behavior of the training and test loss, whereas sub-figures (b) and (d-g) depict the per epoch bias trajectory.

## A.7 The trajectory of higher layer biases

Our last experiment examines the training dynamics associated with the biased of higher layers. We focus on Task 1 and replicate the experiment described in Section 6, but now we track the normalized bias trajectory length of all biases and optimize $W_1$ freely.

Figure A.7 reports the obtained results. It can be observed that the bias dynamics correlate with task complexity for the first three layers, while being uncorrelated for the last two. To interpret these results, we recall that in the proof of Lemma 1 the trajectory length of $b_l$ relates to the Lipschitz constant of subnetwork $f_{d\leftarrow l+1} = f_d \circ \cdots \circ f_{l+1}$ close to the training data. Then, noticing how the trajectory length decays by almost an order of magnitude at each layer, we may infer that the Lipschitz constant of $f_{d\leftarrow l+1}$ decays quickly with $l$. The latter implies that the NN predominantly employs the first few layers to solve the task, whereas the last two layers implement very simple functions.

## B Deferred technical arguments

### B.1 A simple Lemma

**Lemma 1.** *Let $f^{(t)}$ be a $d$-layer NN at the $t$-th SGD iteration, denote by $\boldsymbol{x}^{(t)} \in X$ the point of the training set sampled at that iteration, and set*

$$\epsilon_{f^{(t)}}(\boldsymbol{x}, y) := \left| \frac{\partial \ell(\hat{y}, y)}{\partial \hat{y}} \right|_{\hat{y}=f^{(t)}(\boldsymbol{x})}. \tag{1}$$

*The Lipschitz constant of $f^{(t)}$ at $\mathcal{R}_{\boldsymbol{x}^{(t)}}$ is*

$$\frac{\|\boldsymbol{b}_1^{(t+1)} - \boldsymbol{b}_1^{(t)}\|_2}{\alpha_t \cdot \epsilon_{f^{(t)}}(\boldsymbol{x}^{(t)}, y^{(t)})} \cdot \sigma_n(\boldsymbol{W}_1^{(t)}) \leq \lambda_{f^{(t)}}(\mathcal{R}_{\boldsymbol{x}^{(t)}}) \leq \frac{\|\boldsymbol{b}_1^{(t+1)} - \boldsymbol{b}_1^{(t)}\|_2}{\alpha_t \cdot \epsilon_{f^{(t)}}(\boldsymbol{x}^{(t)}, y^{(t)})} \cdot \sigma_1(\boldsymbol{W}_1^{(t)}),$$

*where $\sigma_1(\boldsymbol{W}_1^{(t)}) \geq \cdots \geq \sigma_n(\boldsymbol{W}_1^{(t)}) > 0$ are the singular values of $\boldsymbol{W}_1^{(t)}$.*

Let us start with some basics. By the chain rule, we have

$$\frac{\partial \ell(f(\boldsymbol{x}^{(t)}, \boldsymbol{w}^{(t)}), y^{(t)})}{\partial \boldsymbol{w}^{(t)}} = \frac{\partial \ell(\hat{y}, y^{(t)})}{\partial \hat{y}} \cdot \frac{\partial f(\boldsymbol{x}^{(t)}, \boldsymbol{w}^{(t)})}{\partial \boldsymbol{w}^{(t)}}$$

with $\hat{y} = f(\boldsymbol{x}^{(t)}, \boldsymbol{w}^{(t)})$, whereas the gradient w.r.t. the bias of the $\ell$-th layer is given by

$$\left(\frac{\partial f(\boldsymbol{x}^{(t)}, \boldsymbol{w}^{(t)})}{\partial \boldsymbol{b}_l^{(t)}}\right)^{\top} = \boldsymbol{S}_d^{(t)}(\boldsymbol{x}^{(t)})\boldsymbol{W}_d^{(t)} \cdots \boldsymbol{S}_{l+1}^{(t)}(\boldsymbol{x}^{(t)})\boldsymbol{W}_{l+1}^{(t)}\boldsymbol{S}_l^{(t)}(\boldsymbol{x}^{(t)}).$$

Note that the above equation abuses notation for the last layer activation $\boldsymbol{S}_d^{(t)}(\boldsymbol{x}^{(t)})$. Specifically, depending on whether we are using an identity or sigmoid activation function in the last layer, we set

$$\boldsymbol{S}_d^{(t)}(\boldsymbol{x}^{(t)}) = 1 \quad \text{or} \quad \boldsymbol{S}_d^{(t)}(\boldsymbol{x}^{(t)}) = \psi\left(\boldsymbol{W}_d^{(t)}\left(f_{d-1}^{(t)} \circ \cdots \circ f_1^{(t)}(\boldsymbol{x}^{(t)})\right) + \boldsymbol{b}_d^{(t)}\right), \qquad (2)$$

where $\psi(z) = \frac{\partial \rho_d(z)}{\partial z} = \frac{1}{1+e^{-z}} \cdot \left(1 - \frac{1}{1+e^{-z}}\right)$.

**Part 1.** Define the following shorthand notation:

$$\left\|\left(\frac{\partial \ell(f(\boldsymbol{x}^{(t)}, \boldsymbol{w}^{(t)}), y^{(t)})}{\partial \boldsymbol{b}_l}\right)^{\top}\right\|_2 = \beta_l(\boldsymbol{x}^{(t)})$$

It follows from definition that

$$\beta_l(\boldsymbol{x}^{(t)}) = \left\|\frac{\partial \ell(o, y^{(t)})}{\partial o} \boldsymbol{S}_d^{(t)}(\boldsymbol{x}^{(t)})\boldsymbol{W}_d^{(t)} \cdots \boldsymbol{S}_l^{(t)}(\boldsymbol{x}^{(t)})\boldsymbol{W}_{l+1}^{(t)}\boldsymbol{S}_l(\boldsymbol{x}^{(t)})\right\|_2$$

$$= \left|\frac{\partial \ell(o, y^{(t)})}{\partial o}\right| \left\|\left(\prod_{i=d}^{l+1} \boldsymbol{S}_i^{(t)}(\boldsymbol{x}^{(t)})\boldsymbol{W}_i^{(t)}\right)\boldsymbol{S}_l^{(t)}(\boldsymbol{x}^{(t)})\right\|_2$$

or equivalently,

$$\left\|\left(\prod_{i=d}^{l+1} \boldsymbol{S}_i^{(t)}(\boldsymbol{x}^{(t)})\boldsymbol{W}_i^{(t)}\right)\boldsymbol{S}_l^{(t)}(\boldsymbol{x}^{(t)})\right\|_2 = \beta_l(\boldsymbol{x}^{(t)})\left|\frac{\partial \ell(o, y^{(t)})}{\partial o}\right|^{-1} = \frac{\beta_l(\boldsymbol{x}^{(t)})}{\epsilon_{f^{(t)}}(\boldsymbol{x}^{(t)}, y^{(t)})}. \qquad (3)$$

**Part 2.** We are interested in upper bounding the Lipschitz constant $\lambda_{f^{(t)}}$ of the NN close to the training data $X$.

First observe that $f(\boldsymbol{x}, \boldsymbol{w}) = f_{d\leftarrow 2} \circ f_1(\boldsymbol{x})$, where we set

$$f_{d\leftarrow 2}(\boldsymbol{x}, \boldsymbol{w}) = f_d \circ f_{d-1} \circ \cdots \circ f_2(\boldsymbol{x}, \boldsymbol{w})$$

Let $\mathcal{R}_{\boldsymbol{x}^{(t)}}$ be the region associated with point $\boldsymbol{x}^{(t)}$ and $\mathcal{R}_{f_1^{(t)}(\boldsymbol{x}^{(t)})}$ the region of the NN $f_{d\leftarrow 2}^{(t)}$ associated with point $f_1^{(t)}(\boldsymbol{x}^{(t)})$. The Lipschitz constants of $f_{d\leftarrow 2}^{(t)}$ and $f^{(t)}$ are related as follows:

$$\lambda_{f_{d\leftarrow 2}^{(t)}}(\mathcal{R}_{f_1^{(t)}(\boldsymbol{x}^{(t)})}) \cdot \sigma_n(\boldsymbol{W}_1^{(t)}) \le \lambda_{f^{(t)}}(\mathcal{R}_{\boldsymbol{x}^{(t)}}) \le \lambda_{f_{d\leftarrow 2}^{(t)}}(\mathcal{R}_{f_1^{(t)}(\boldsymbol{x}^{(t)})}) \cdot \sigma_1(\boldsymbol{W}_1^{(t)}), \qquad (4)$$

whereas

$$\lambda_{f_{d\leftarrow 2}}(\mathcal{R}_{f_1^{(t)}(\boldsymbol{x}^{(t)})}) = \left\|\left(\prod_{l=d}^{2} \boldsymbol{S}_l^{(t)}(\boldsymbol{x}^{(t)})\boldsymbol{W}_l^{(t)}\right)\boldsymbol{S}_1^{(t)}(\boldsymbol{x}^{(t)})\right\|_2 = \frac{\beta_1(\boldsymbol{x}^{(t)})}{\epsilon_{f^{(t)}}(\boldsymbol{x}^{(t)}, y^{(t)})}. \qquad (5)$$

Combining (4) with (5), we obtain

$$\frac{\beta_1(\boldsymbol{x}^{(t)})}{\epsilon_{f^{(t)}}(\boldsymbol{x}^{(t)}, y^{(t)})} \cdot \sigma_n(\boldsymbol{W}_1^{(t)}) \le \lambda_{f^{(t)}}(\mathcal{R}_{\boldsymbol{x}^{(t)}}) \le \frac{\beta_1(\boldsymbol{x}^{(t)})}{\epsilon_{f^{(t)}}(\boldsymbol{x}^{(t)}, y^{(t)})} \cdot \sigma_1(\boldsymbol{W}_1^{(t)})$$

**Part 3.** Re-organizing the SGD expression and taking the norm, we have

$$\beta_l(\boldsymbol{x}^{(t)}) = \left\|\left(\frac{\partial \ell(f(\boldsymbol{x}^{(t)}, \boldsymbol{w}^{(t-1)}), y^{(t)})}{\partial \boldsymbol{b}_l^{(t)}}\right)^{\top}\right\|_2 = \frac{1}{\alpha_t}\|\boldsymbol{b}_l^{(t+1)} - \boldsymbol{b}_l^{(t)}\|_2.$$

implying also

$$\frac{\|\boldsymbol{b}_1^{(t+1)} - \boldsymbol{b}_1^{(t)}\|_2}{\alpha_t \cdot \epsilon_{f^{(t)}}(\boldsymbol{x}^{(t)}, y^{(t)})} \cdot \sigma_n(\boldsymbol{W}_1^{(t)}) \le \lambda_{f^{(t)}}(\mathcal{R}_{\boldsymbol{x}^{(t)}}) \le \frac{\|\boldsymbol{b}_1^{(t+1)} - \boldsymbol{b}_1^{(t)}\|_2}{\alpha_t \cdot \epsilon_{f^{(t)}}(\boldsymbol{x}^{(t)}, y^{(t)})} \cdot \sigma_1(\boldsymbol{W}_1^{(t)}),$$

as claimed.

## B.2   Proof of Theorem 1

The proof of the theorem follows directly from Lemma 1 by summing over the training trajectory:

$$\sum_{t \in T} \frac{\lambda_{f^{(t)}}(\mathcal{R}_{\boldsymbol{x}})}{\sigma_n(\boldsymbol{W}_1^{(t)})} \overset{\text{Lemma 1}}{\geq} \sum_{t \in T} \frac{\|\boldsymbol{b}_1^{(t+1)} - \boldsymbol{b}_1^{(t)}\|_2}{\alpha_t \, \epsilon_{f^{(t)}}(\boldsymbol{x}^{(t)}, y^{(t)})} \overset{\text{Lemma 1}}{\geq} \sum_{t \in T} \frac{\lambda_{f^{(t)}}(\mathcal{R}_{\boldsymbol{x}})}{\sigma_1(\boldsymbol{W}_1^{(t)})}.$$

## B.3   Proof of Corollary 1

For any point $\boldsymbol{x}$ with label $y$ and iteration $t \in T$, we say that condition $c_t(\boldsymbol{x})$ holds if

$$\|\boldsymbol{b}_1^{(t+1)} - \boldsymbol{b}_1^{(t)}\| \leq \varphi \, \alpha_t \, \epsilon_{f^{(t)}}(\boldsymbol{x}, y)$$

The above is the same condition as in the corollary statement but applied to an arbitrary point $\boldsymbol{x}$.

Write $\kappa_t$ to refer to the number of training points $\boldsymbol{x}_i \in X$ for which $c_t(\boldsymbol{x}_i)$ holds: clearly, $\kappa_t \in [0, N]$, where $N$ is the size of the training set.

We also suppose that there are $\xi$ iterations within $T$ for which $\kappa_t < N$: these are the iterations where the Lipschitz constant is larger than $\beta\varphi$ for at least one point in the training set.

Since we sample $\boldsymbol{x}^{(t)}$ i.i.d. from $X$, the probability that $c_t(\boldsymbol{x}^{(t)})$ is satisfied for every $t \in T$ is at most

$$\prod_{t \in T} \frac{\kappa_t}{N} \leq \left(\frac{N-1}{N}\right)^{\xi} = \left(1 - \frac{1}{N}\right)^{\xi} \leq e^{-\xi/N}.$$

By noting that above corresponds to the probability that the NN is $(\tau, \varphi)$-steady, we deduce that $\xi$ cannot grow with $|T|$ (otherwise, the probability that the NN is $(\tau, \varphi)$-steady would become zero as $|T| \to \infty$, which contradicts the corollary assumptions).

To complete the derivation, we note that if we select the iteration $t$ at random from $T$, the probability that there will be some $\boldsymbol{x}_i \in X$ for which $c_t(\boldsymbol{x}_i)$ is not satisfied is $\xi/|T| = O(1/|T|)$, which converges to 0 as $|T|$ grows.

## B.4   Proof of Corollary 2

We consider the interval $T = \{t_1 + 1, \dots, t_2\}$ and fix

$$\boldsymbol{b}_1 = \operatorname{argmin}_{\boldsymbol{b} \in \mathbb{R}^n} \sum_{t \in T} \|\boldsymbol{b}_1^{(t)} - \boldsymbol{b}\|_2^2 = \sum_{t \in T} \frac{\boldsymbol{b}_1^{(t)}}{|T|}$$

to be the average bias. Working as in the proof of Theorem 1, we deduce

$$\sum_{t \in T} \frac{\|\boldsymbol{b}_1^{(t+1)} - \boldsymbol{b}_1^{(t)}\|_2}{\epsilon_{f^{(t)}}(\boldsymbol{x}^{(t)}, y^{(t)})} \geq \sum_{t \in T} \frac{\alpha_t \lambda_{f^{(t)}}(\mathcal{R}_{\boldsymbol{x}^{(t)}})}{\sigma_1(\boldsymbol{W}_1^{(t)})}.$$

We then proceed to upper bound the trajectory length studied in Theorem 1 in terms of the (empirical) variance of the bias:

$$\left(\sum_{t \in T} \frac{\|\boldsymbol{b}_1^{(t+1)} - \boldsymbol{b}_1^{(t)}\|_2}{\epsilon_{f^{(t)}}(\boldsymbol{x}^{(t)}, y^{(t)})}\right)^2 \leq \left(\sum_{t \in T} \frac{\|\boldsymbol{b}_1^{(t+1)} - \boldsymbol{b}_1\|_2 + \|\boldsymbol{b}_1^{(t)} - \boldsymbol{b}_1\|_2}{\epsilon_{f^{(t)}}(\boldsymbol{x}^{(t)}, y^{(t)})}\right)^2$$

$$\leq \left(\sum_{t \in T} \epsilon_{f^{(t)}}(\boldsymbol{x}^{(t)}, y^{(t)})^{-2}\right) \left(\sum_{t \in T} \left(\|\boldsymbol{b}_1^{(t+1)} - \boldsymbol{b}_1\|_2 + \|\boldsymbol{b}_1^{(t)} - \boldsymbol{b}_1\|_2\right)^2\right)$$

$$\text{(From Cauchy's inequality)}$$

$$\leq \sum_{t \in T} \epsilon_{f^{(t)}}(\boldsymbol{x}^{(t)}, y^{(t)})^{-2} \sum_{t \in T} 2\left(\|\boldsymbol{b}_1^{(t+1)} - \boldsymbol{b}_1\|_2^2 + \|\boldsymbol{b}_1^{(t)} - \boldsymbol{b}_1\|_2^2\right)$$

$$\text{(since } (a+b)^2 \leq 2(a^2 + b^2))$$

$$\leq 4 \sum_{t \in T} \epsilon_{f^{(t)}}(\boldsymbol{x}^{(t)}, y^{(t)})^{-2} \sum_{t=t_1}^{t_2} \|\boldsymbol{b}_1^{(t)} - \boldsymbol{b}_1\|_2^2$$

or, equivalently,

$$\sum_{t=t_1}^{t_2} \frac{\|\boldsymbol{b}_1^{(t)} - \boldsymbol{b}_1\|_2^2}{|T|} \geq \frac{1}{4|T|} \left( \sum_{t \in T} \frac{\|\boldsymbol{b}_1^{(t+1)} - \boldsymbol{b}_1^{(t)}\|_2}{\epsilon_{f^{(t)}}(\boldsymbol{x}^{(t)}, y^{(t)})} \right)^2 \frac{1}{\sum_{t \in T} \epsilon_{f^{(t)}}(\boldsymbol{x}^{(t)}, y^{(t)})^{-2}} \tag{6}$$

Thus, we have

$$\sum_{t=t_1}^{t_2} \frac{\|\boldsymbol{b}_1^{(t)} - \boldsymbol{b}_1\|_2^2}{|T|} \geq 0.25 \left( \underset{t=t_1}{\overset{t_2}{\text{avg}}} \frac{\alpha_t \lambda_{f^{(t)}}(\mathcal{R}_{\boldsymbol{x}^{(t)}})}{\sigma_1(\boldsymbol{W}_1^{(t)})} \right)^2 \frac{|T|}{\sum_{t \in T} \frac{1}{\epsilon_{f^{(t)}}(\boldsymbol{x}^{(t)}, y^{(t)})^2}}. \tag{7}$$

The proof concludes by noticing that the right-most term corresponds to a harmonic mean of $\epsilon_{f^{(t)}}(\boldsymbol{x}^{(t)}, y^{(t)})^2$ over $t \in T$.

## B.5  Proof of Corollary 3

Suppose that we train our NN for $\tau$ iterations and set $T = \{0, \ldots, \tau-1\}$. The distance to initialization is bounded by

$$\|\boldsymbol{b}_1^{(\tau)} - \boldsymbol{b}_1^{(0)}\|_2 \leq \sum_{t \in T} \|\boldsymbol{b}_1^{(t+1)} - \boldsymbol{b}_1^{(t)}\|_2 = \sum_{t \in T} \frac{\|\boldsymbol{b}_1^{(t+1)} - \boldsymbol{b}_1^{(t)}\|_2}{\epsilon_{f^{(t)}}(\boldsymbol{x}^{(t)}, y^{(t)})} \epsilon_{f^{(t)}}(\boldsymbol{x}^{(t)}, y^{(t)}).$$

We obtain the final expression by arguing as in the proof of Theorem 1 to write:

$$\sum_{t \in T} \frac{\|\boldsymbol{b}_1^{(t+1)} - \boldsymbol{b}_1^{(t)}\|_2}{\epsilon_{f^{(t)}}(\boldsymbol{x}^{(t)}, y^{(t)})} \epsilon_{f^{(t)}}(\boldsymbol{x}^{(t)}, y^{(t)}) \leq \sum_{t \in T} \frac{\alpha_t \, \epsilon_{f^{(t)}}(\boldsymbol{x}^{(t)}, y^{(t)}) \lambda_{f^{(t)}}(\mathcal{R}_{\boldsymbol{x}^{(t)}})}{\sigma_n(\boldsymbol{W}_1^{(t)})}.$$

## B.6  Proof of Theorem 2

We will start by proving the following Lemma:

**Lemma 2.** *Let $\mathcal{R}_{\boldsymbol{x}}$ be a linear region of $f^{(t)}$ and suppose that there exists a vector $\boldsymbol{a} \in \mathbb{R}^{|T|}$ such that*

$$\prod_{l=d-1}^{1} [\boldsymbol{S}_l^{(t)}(\boldsymbol{x})](i_l, i_l) = \sum_{k \in T} \boldsymbol{a}(k) \cdot \prod_{l=d-1}^{1} [\boldsymbol{S}_l^{(k)}(\boldsymbol{x}^{(k)})](i_l, i_l) \tag{8}$$

*for all indices $\{i_l\}_{l=1,\cdots,d-1}$, with $i_l \in \{1, \ldots, n_l\}$. Then, $f^{(t)}$ is Lipschitz continuous within $\mathcal{R}_{\boldsymbol{x}}$ and its Lipschitz constant is at most*

$$\lambda_{f^{(t)}}(\mathcal{R}_{\boldsymbol{x}}) \leq (1 + \gamma) \sum_{k \in T} |\boldsymbol{a}(k)| \frac{|\boldsymbol{S}_d^{(t)}(\boldsymbol{x})|}{|\boldsymbol{S}_d^{(t)}(\boldsymbol{x}^{(k)})|} \lambda_{f^{(k)}}(\mathcal{R}_{\boldsymbol{x}^{(k)}}),$$

*with*

$$\boldsymbol{S}_d^{(t)}(\boldsymbol{z}) = \frac{\partial \rho_d(x)}{\partial x} \bigg|_{x = \boldsymbol{W}_d^{(t)} \left( f_{d-1}^{(t)} \circ \cdots \circ f_1^{(t)}(\boldsymbol{z}) \right) + \boldsymbol{b}_d^{(t)}}. \tag{9}$$

*Proof.* The gradient of a network $f^{(t)}$ at at point $\boldsymbol{x}$ is simply

$$\boldsymbol{\nabla} f^{(t)}(\boldsymbol{x}) = \prod_{l=d}^{1} \boldsymbol{S}_l^{(t)}(\boldsymbol{x}) \boldsymbol{W}_l^{(t)} = \boldsymbol{S}_d^{(t)}(\boldsymbol{x}) \boldsymbol{W}_d^{(t)} \overbrace{\prod_{l=d-1}^{1} \boldsymbol{S}_l^{(t)}(\boldsymbol{x}) \boldsymbol{W}_l^{(t)}}^{q^{(t)}(\boldsymbol{x})}.$$

Term $q^{(t)}(\boldsymbol{x})$ can be expanded as follows:

$$q^{(t)}(\boldsymbol{x}) = \sum_{i_{d-1}=1}^{n_d} \boldsymbol{W}_d^{(t)}(1, i_{d-1}) \left[ \prod_{l=d-1}^{1} \boldsymbol{S}_l^{(t)}(\boldsymbol{x}) \boldsymbol{W}_l^{(t)} \right] (i_{d-1}, :)$$

$$= \cdots$$

$$= \sum_{i_{d-1}=1}^{n_{d-1}} \cdots \sum_{i_1=1}^{n_1} \boldsymbol{W}_d^{(t)}(1, i_{d-1}) \cdots [\boldsymbol{S}_1^{(t)}(\boldsymbol{x})](i_1, i_1) \, \boldsymbol{W}_1^{(t)}(i_1, :).$$

Thus, under the main Lemma condition it is also true that

$$q^{(t)}(\boldsymbol{x}) = \sum_{k \in T} \boldsymbol{a}(k) \cdot \left( \sum_{i_{d-1}=1}^{n_{d-1}} \cdots \sum_{i_1=1}^{n_1} \boldsymbol{W}_d^{(t)}(1, i_{d-1}) \cdots [\boldsymbol{S}_1^{(k)}(\boldsymbol{x}^{(k)})](i_1, i_1) \, \boldsymbol{W}_1^{(t)}(i_1, :) \right)$$

$$= \sum_{k \in T} \boldsymbol{a}(k) \cdot \left( \sum_{i_{d-1}=1}^{n_{d-1}} \cdots \sum_{i_1=1}^{n_1} \boldsymbol{W}_d^{(t)}(1, i_{d-1}) \cdots [\boldsymbol{S}_1^{(t)}(\boldsymbol{x}^{(k)})](i_1, i_1) \, \boldsymbol{W}_1^{(t)}(i_1, :) \right)$$

$$= \sum_{k \in T} \boldsymbol{a}(k) \cdot q^{(t)}(\boldsymbol{x}^{(k)}),$$

Note that in the second step above we have used Assumption 1 to argue that the training point activation patterns do not change within $T$.

The above analysis implies that

$$\lambda_{f^{(t)}}(\mathcal{R}_{\boldsymbol{x}}) = |\boldsymbol{S}_d^{(t)}(\boldsymbol{x})| \|q^{(t)}(\boldsymbol{x})\|_2 = |\boldsymbol{S}_d^{(t)}(\boldsymbol{x})| \| \sum_{k \in T} \boldsymbol{a}(k) \cdot q^{(t)}(\boldsymbol{x}^{(k)}) \|_2$$

$$\leq |\boldsymbol{S}_d^{(t)}(\boldsymbol{x})| \sum_{k \in T} \|\boldsymbol{a}(k) \cdot q^{(t)}(\boldsymbol{x}^{(k)})\|_2$$

$$= \sum_{k \in T} |\boldsymbol{a}(k)| \cdot \frac{|\boldsymbol{S}_d^{(t)}(\boldsymbol{x})|}{|\boldsymbol{S}_d^{(t)}(\boldsymbol{x}^{(k)})|} \cdot \lambda_{f^{(t)}}(\mathcal{R}_{\boldsymbol{x}^{(k)}})$$

$$\leq (1 + \gamma) \sum_{k \in T} |\boldsymbol{a}(k)| \cdot \frac{|\boldsymbol{S}_d^{(t)}(\boldsymbol{x})|}{|\boldsymbol{S}_d^{(t)}(\boldsymbol{x}^{(k)})|} \cdot \lambda_{f^{(k)}}(\mathcal{R}_{\boldsymbol{x}^{(k)}})$$

with the 3rd step being true due to the triangle inequality and the 5th follows from Assumption 1. □

The proof continues by realizing that, for every index set $i_{d-1}, \cdots, i_1$ there exists an entry $i$ such that

$$[ \bigotimes_{l=d-1}^{1} \boldsymbol{S}_l^{(t)}(\boldsymbol{x})](i, i) = \prod_{l=d-1}^{1} [\boldsymbol{S}_l^{(t)}(\boldsymbol{x})](i_l, i_l).$$

Therefore, condition (8) is equivalent to asserting that

$$\boldsymbol{s}_t(\boldsymbol{x}) = \bigotimes_{l=d-1}^{1} \text{diag}\left( \boldsymbol{S}_l^{(t)}(\boldsymbol{x}) \right) = \sum_{k \in T} \boldsymbol{a}(k) \cdot \bigotimes_{l=d-1}^{1} \text{diag}\left( \boldsymbol{S}_l^{(k)}(\boldsymbol{x}^{(k)}) \right)$$

$$= \sum_{k \in T} \boldsymbol{a}(k) \cdot \boldsymbol{s}_k(\boldsymbol{x}^{(k)}) = \boldsymbol{S}_T \, \boldsymbol{a}.$$

Let us focus on $|\boldsymbol{S}_d^{(t)}(\boldsymbol{x})|/|\boldsymbol{S}_d^{(t)}(\boldsymbol{x}^{(k)})|$. When there is no activation in the last layer, the term is trivially $\xi = 1$. We next derive an upper bound to also account for the sigmoid activation: To do this, set $z = \boldsymbol{W}_d^{(t)}\left( f_{d-1}^{(t)} \circ \cdots \circ f_1^{(t)}(\boldsymbol{x}) \right) + \boldsymbol{b}_d^{(t)}$ such that

$$\boldsymbol{S}_d^{(t)}(\boldsymbol{x}) = \psi(z) \quad \text{with} \quad \psi(z) = \frac{\partial \rho_d(z)}{\partial z} = \frac{1}{1 + e^{-z}} \cdot \left( 1 - \frac{1}{1 + e^{-z}} \right)$$

Function $\psi$ takes its maximum value for $z = 0$, with $\psi(z) \leq \psi(0) = 0.25$. We notice that $\psi$ is symmetric around 0 and monotonically decreasing on either side. Its minimum is thus given when $|z|$ is as large as possible. However, since our classifier's output is bounded in $f^{(t)}(\boldsymbol{x}) \in [\mu_T, 1 - \mu_T]$ for all points seen within $T$, we have $|z| \leq \log(1/\mu_T - 1)$ and thus

$$|\boldsymbol{S}_d^{(t)}(\boldsymbol{x}^{(k)})| \geq \psi(\log(1/\mu_T - 1)) = \mu_T(1 - \mu_T).$$

All in all, we get $|S_d^{(t)}(\boldsymbol{x})|/|S_d^{(t)}(\boldsymbol{x}^{(k)})| \leq 0.25/(\mu_T(1 - \mu_T)) = \xi$.

We then rely on Lemma 1 to upper bound each local Lipschitz constant in terms of the bias update:

$$\lambda_{f^{(k)}}(\mathcal{R}_{\boldsymbol{x}^{(k)}}) \leq \frac{\|\boldsymbol{b}_1^{(k+1)} - \boldsymbol{b}_1^{(k)}\|_2}{\alpha_k \, \epsilon_{f^{(k)}}(\boldsymbol{x}^{(k)}, y^{(k)})} \, \sigma_1(\boldsymbol{W}_1^{(k)})_2 \leq \beta \, \frac{\|\boldsymbol{b}_1^{(k+1)} - \boldsymbol{b}_1^{(k)}\|_2}{\alpha_k \, \epsilon_{f^{(k)}}(\boldsymbol{x}^{(k)}, y^{(k)})}, \tag{10}$$

matching the claim of the theorem.

## B.7 Proof of Theorem 3

We repeat the theorem statement here for easy reference:

**Theorem 3.** *Let $f^{(t)}$ be a depth $d$ NN with ReLU activations being trained with SGD, a BCE loss and ½-Dropout.*

*Suppose that $f^{(t)}$ is $(\tau, \varphi)$-steady and that for every $t \geq \tau$ the following hold: (a) Assumption 1, (b) $s_t(\boldsymbol{x}) \leq \sum_{i=1}^N s_t(\boldsymbol{x}_i)$ for every $\boldsymbol{x} \in \mathcal{X}$, (c) $\sigma_1(\boldsymbol{W}_1^{(t)}) \leq \beta$, and (d) $f^{(t)}(\boldsymbol{x}^{(t)}) \in [\mu, 1 - \mu]$.*

*Define*

$$r_t(X) = \frac{\min_{i=1}^N |1 - 2f^{(t)}(\boldsymbol{x}_i)|}{c \, \varphi \, \log\left(\sum_{l=1}^{d-1} n_l\right)} \quad and \quad c = \frac{(1 + \gamma) \, \beta \, (1 + o(1))}{\mu \, (1 - \mu) \, p_{min}},$$

*where $p_{min} = \min_{l < d, i \leq n_l, t \geq \tau}[\text{avg}_{\boldsymbol{x} \in X} \, diag(S_l^{(t)}(\boldsymbol{x}))]_i > 0$ is the minimum frequency that any neuron is active before Dropout is applied.*

*For any $\delta > 0$, with probability at least $1 - \delta$ over the Dropout and the training set sampling, the generalization error is at most*

$$\left|er_t^{emp} - er_t^{exp}\right| \leq \sqrt{\frac{4 \log(2) \, \mathcal{N}(\mathcal{X}; \ell_2, r(X)) + 2 \log(1/\delta)}{N}},$$

*where $\mathcal{N}(\mathcal{X}; \ell_2, r)$ is the minimal number of $\ell_2$-balls of radius $r$ needed to cover $\mathcal{X}$.*

The proof consists of two parts. First, Lemma 3 provides a bound on the *global* Lipschitz constant of a NN trained with Dropout as a function of the bias updates observed during a sufficiently long training. Then, Lemma 4 uses techniques from the robustness framework [2, 3] to derive a generalization bound.

### B.7.1 The global Lipschitz constant

We prove the following:

**Lemma 3.** *In the setting of Theorem 2, suppose that the network is trained using ½-Dropout and denote by $\boldsymbol{p}_l = \text{avg}_{\boldsymbol{x} \in X} \, diag\left(S_l^{(t)}(\boldsymbol{x})\right)$ the probability that the neurons in layer $l$ are active (before Dropout is applied). The global Lipschitz constant of $f^{(t)}$ is with high probability*

$$\lambda_{f^{(t)}} \leq c \, \log\left(\sum_{l=1}^{d-1} n_l\right) \|\boldsymbol{\varphi}_T\|_\infty := \lambda_{f^{(t)}}^{steady}$$

*for $c = (1 + \gamma) \, \beta \, \xi \, (1 + o(1))/p_{min}$, whenever $|T| = \tilde{\Omega}\left(\frac{p_{avg}}{p_{min}^2} \sum_{l=1}^{d-1} n_l\right)$, with $p_{min}$ and $p_{avg}$ being the minimum and average entry over all $\boldsymbol{p}_l$, respectively.*

The inequality provided above is unexpectedly tight: combining $\lambda_{f^{(t)}} \geq \lambda_{f^{(t)}}(\mathcal{R}_{\boldsymbol{x}^{(t)}})$ with Lemma 1 we can deduce that

$$\lambda_{f^{(t)}} \leq \lambda_{f^{(t)}}^{steady} \leq \lambda_{f^{(t)}} \, O(\log(dn)),$$

where we have assumed that $c/\sigma_n(\boldsymbol{W}_1) = O(1)$ and $n_l = n$ for all $l < d$.

*Proof.* Let $\boldsymbol{x}$ be a point within a region where $f^{(t)}$ assumes its maximum gradient norm.

The activation $\tilde{\boldsymbol{S}}_T(:, t) = \tilde{\boldsymbol{s}}_t(\boldsymbol{x}^{(t)})$ at the $t$-th SGD iteration is obtained by a two step procedure:

1. Sample a point $\boldsymbol{x}^{(t)}$ from $X$ with replacement. Let $\boldsymbol{s}_t(\boldsymbol{x}^{(t)}) = \bigotimes_{l=d-1}^{1} \boldsymbol{s}_{t,l}(\boldsymbol{x}^{(t)})$ be its activation pattern (before dropout), where $\boldsymbol{s}_{t,l}(\boldsymbol{x}^{(t)}) := \mathrm{diag}\left( \boldsymbol{S}_l^{(t)}(\boldsymbol{x}^{(t)}) \right)$.

2. Construct $\tilde{\boldsymbol{s}}_t(\boldsymbol{x}^{(t)})$ by setting each neuron activation to zero with probability 0.5. Specifically, $\tilde{\boldsymbol{s}}_t(\boldsymbol{x}^{(t)}) = \bigotimes_{l=d-1}^{1} (\boldsymbol{z}_l \circ \boldsymbol{s}_{t,l}(\boldsymbol{x}^{(t)}))$, where $\boldsymbol{z}_l \in \{0,1\}^{n_l}$ is a random binary vector.

Let $\boldsymbol{S}$ be a binary matrix containing neuron activations as columns. We introduce the following definitions:

- We call $\boldsymbol{S}$ a *covering set* if $\boldsymbol{S}\boldsymbol{1} \geq \boldsymbol{1}$ with the inequality taken element-wise.
- We call $\boldsymbol{S}$ a *basis* of $\boldsymbol{s}_t(\boldsymbol{x})$ if $\boldsymbol{S}\boldsymbol{1} = \boldsymbol{s}_t(\boldsymbol{x})$.

Our proof hinges on two observations:

*Observation 1.* Every basis yields a bound on the Lipschitz constant of $f^{(t)}$ (this can be seen from the proof of Theorem 2). Specifically, for any $k$ training points $\boldsymbol{x}_1, \ldots, \boldsymbol{x}_k$ whose activations $\boldsymbol{S} = [\boldsymbol{s}_t(\boldsymbol{x}_1), \ldots, \boldsymbol{s}_t(\boldsymbol{x}_k)]$ is a basis of $\boldsymbol{s}_t(\boldsymbol{x})$, we have

$$\lambda_{f^{(t)}}(\mathcal{R}_{\boldsymbol{x}}) \leq \xi \sum_{i=1}^{k} \lambda_{f^{(t)}}(\mathcal{R}_{\boldsymbol{x}_i}) \leq k\,\xi\, \max_{i=1}^{k} \lambda_{f^{(t)}}(\mathcal{R}_{\boldsymbol{x}_i}).$$

where $\xi \geq \dfrac{|\boldsymbol{S}_d^{(t)}(\boldsymbol{x})|}{|\boldsymbol{S}_d^{(t)}(\boldsymbol{x}_i)|}$ accounts for the sigmoid.

Thus, if we don't use dropout and within the columns of $\boldsymbol{S}_T$ there exist $k$ that form a basis of $\boldsymbol{s}_t(\boldsymbol{x})$, then this also implies that the global Lipschitz constant will be bounded by

$$\lambda_{f^{(t)}}(\mathcal{R}_{\boldsymbol{x}}) \leq k\beta(1+\gamma)\xi \max_{t \in T} \frac{\|\boldsymbol{b}_1^{(t+1)} - \boldsymbol{b}_1^{(t)}\|_2}{\alpha_t \epsilon_{f^{(t)}}(\boldsymbol{x}^{(t)}, y^{(t)})},$$

where, in an identical fashion to Theorem 2, the $1+\gamma$ factor is added due to Assumption 1 in order to account for $f^{(t)}$ not having completely converged, and we have also used Lemma 1 and the uniform bound $\|\boldsymbol{W}_1^{(t)}\|_2 \leq \beta$.

*Observation 2.* Let us consider the effect of Dropout. Suppose that $\boldsymbol{S}_T$ does not contain a basis of $\boldsymbol{s}_t(\boldsymbol{x}) = \bigotimes_{l=d-1}^{1} \boldsymbol{s}_{t,l}(\boldsymbol{x})$, but there exist a set of columns $\boldsymbol{S}$ that is a covering set (as we will see, this is a much easier condition to satisfy). Denote by $\tilde{\boldsymbol{S}}$ the same matrix after the Dropout sampling. Then, with some strictly positive probability, $\tilde{\boldsymbol{S}}$ can become a basis.

**Claim 1.** *For any $k$ training points $\boldsymbol{x}_1, \ldots, \boldsymbol{x}_k$ whose activations $\boldsymbol{S} = [\boldsymbol{s}_t(\boldsymbol{x}_1), \ldots, \boldsymbol{s}_t(\boldsymbol{x}_k)]$ form a covering set, there must exist $\boldsymbol{Q} = [\boldsymbol{q}_1, \ldots, \boldsymbol{q}_k]$ with $\boldsymbol{q}_i = \bigotimes_{l=d-1}^{1} \boldsymbol{q}_{i,l}$ and $\boldsymbol{q}_{i,l} \leq \boldsymbol{s}_{t,l}(\boldsymbol{x}_i)$ (i.e., that Dropout can sample) such that $\boldsymbol{Q}$ is a basis of $\boldsymbol{s}_t(\boldsymbol{x})$.*

*Proof.* To deduce this fact, we notice that since

$$\sum_{i=1}^{k} \boldsymbol{q}_i = \sum_{i=1}^{k} \bigotimes_{l=d-1}^{1} \boldsymbol{q}_{i,l} = \bigotimes_{l=d-1}^{1} \left( \sum_{i=1}^{k} \boldsymbol{q}_{i,l} \right) \quad \text{and} \quad \boldsymbol{s}_t(\boldsymbol{x}) = \bigotimes_{l=d-1}^{1} \boldsymbol{s}_{t,l}(\boldsymbol{x}),$$

to ensure that $\boldsymbol{Q}$ is a basis we need to show that, for every $l$, there exists $[\boldsymbol{q}_{1,l} \cdots \boldsymbol{q}_{k,l}]$ with $\boldsymbol{q}_{i,l} \leq \boldsymbol{s}_{t,l}(\boldsymbol{x}_i)$ such that $\sum_{i=1}^{k} \boldsymbol{q}_{i,l} = \boldsymbol{s}_{t,l}(\boldsymbol{x})$. The latter can always be satisfied when $\sum_{i=1}^{k} \boldsymbol{s}_{t,l}(\boldsymbol{x}) \geq \boldsymbol{1}$. When $\boldsymbol{S}$ is a covering set we have

$$\boldsymbol{S}\boldsymbol{1} = \sum_{i=1}^{k} \bigotimes_{l=d-1}^{1} \boldsymbol{s}_{t,l}(\boldsymbol{x}_i) = \bigotimes_{l=d-1}^{1} \sum_{i=1}^{k} \boldsymbol{s}_{t,l}(\boldsymbol{x}_i) \geq \boldsymbol{1},$$

which also implies $\sum_{i=1}^{k} \boldsymbol{s}_{t,l}(\boldsymbol{x}_i) \geq \boldsymbol{1}$ as needed. $\qquad \square$

To obtain an upper bound for the Lipschitz constant of $f^{(t)}$, our strategy will entail lower bounding the probability that such a basis of $\boldsymbol{s}_t(\boldsymbol{x})$ will be seen within $T$.

Consider any $k$ training points $\boldsymbol{x}_1, \ldots, \boldsymbol{x}_k$ sampled with replacement from $X$ and let $\boldsymbol{S} = [\boldsymbol{s}_t(\boldsymbol{x}_1), \ldots, \boldsymbol{s}_t(\boldsymbol{x}_k)]$ be the corresponding (random) matrix of neural activations. Further, denote by $p_{\text{cover}}(\boldsymbol{S})$ the probability that $\boldsymbol{S}$ is a covering set.

The probability $p_{\text{basis}}(\boldsymbol{S}_T)$ that $\tilde{\boldsymbol{S}}_T$ contains a basis of $\boldsymbol{s}_t(\boldsymbol{x})$ is given by

$$p_{\text{basis}}(\boldsymbol{S}_T) = 1 - P\left(\tilde{\boldsymbol{S}}_T \text{ does not contain a basis}\right)$$

$$\geq 1 - \prod_{p=1}^{\lfloor \frac{|T|}{k} \rfloor} P\left(\tilde{\boldsymbol{S}}_T(:, (p-1)k+1 : pk) \text{ is not a basis}\right)$$

For every $\tilde{\boldsymbol{S}}_T(:, (p-1)k+1 : pk)$ we have:

$$P\left(\tilde{\boldsymbol{S}}_T(:, (p-1)k+1 : pk) \text{ is a basis}\right) = P\left(\tilde{\boldsymbol{S}} \text{ is a basis}\right)$$

$$= P\left(\tilde{\boldsymbol{S}} \text{ is a basis} \mid \boldsymbol{S} \text{ is a covering set}\right) P(\boldsymbol{S} \text{ is a covering set})$$

$$= P\left(\tilde{\boldsymbol{S}} \text{ is a basis} \mid \boldsymbol{S} \text{ is a covering set}\right) p_{\text{cover}}(\boldsymbol{S}).$$

By Observation 2, if $\boldsymbol{S}$ is a covering set then there must exist $\boldsymbol{q}_i = \bigotimes_{l=d-1}^{1} \boldsymbol{q}_{t,l}(\boldsymbol{x}_i) \leq \boldsymbol{s}_t(\boldsymbol{x}_i) = \bigotimes_{l=d-1}^{1} \boldsymbol{s}_{t,l}(\boldsymbol{x}_i)$, such that $\boldsymbol{Q} = [\boldsymbol{q}_1, \ldots, \boldsymbol{q}_k]$ is a basis of $\boldsymbol{s}_t(\boldsymbol{x})$.

We proceed to compute the probability that the activation pattern sampled by Dropout $\tilde{\boldsymbol{s}}_t(\boldsymbol{x}_i) = \bigotimes_{l=d}^{2}(\boldsymbol{z}_{i,l} \circ \boldsymbol{s}_{t,l}(\boldsymbol{x}_i))$, where $\boldsymbol{z}_{l,t}$ are random binary vectors, is a basis of $\boldsymbol{s}_t(\boldsymbol{x}^{(t)})$ due to $\tilde{\boldsymbol{S}} = \boldsymbol{Q}$:

$$P\left(\tilde{\boldsymbol{S}} \text{ is a basis} \mid \boldsymbol{S} \text{ is a covering set}\right) = P\left(\tilde{\boldsymbol{S}} = \boldsymbol{Q}\right)$$

$$= P(\tilde{\boldsymbol{s}}_t(\boldsymbol{x}_i) = \boldsymbol{q}_i \text{ for } i = 1, \ldots, k)$$

$$= \prod_{i=1}^{k} P(\tilde{\boldsymbol{s}}_t(\boldsymbol{x}_i) = \boldsymbol{q}_i)$$

$$= \prod_{i=1}^{k} P\left(\bigotimes_{l=d-1}^{1} (\boldsymbol{z}_{i,l} \circ \boldsymbol{s}_{t,l}(\boldsymbol{x}_i)) = \bigotimes_{l=d-1}^{1} \boldsymbol{q}_{i,l}\right)$$

$$= \prod_{i=1}^{k} \prod_{l=d-1}^{1} P(\boldsymbol{z}_{i,l} \circ \boldsymbol{s}_{t,l}(\boldsymbol{x}_i) = \boldsymbol{q}_{i,l})$$

$$= \prod_{i=1}^{k} \prod_{l=d-1}^{1} \frac{1}{2^{\|\boldsymbol{s}_{t,l}(\boldsymbol{x}_i)\|_1}} = 2^{-\sum_{i=1}^{k} \sum_{l=d-1}^{1} \|\boldsymbol{s}_{t,l}(\boldsymbol{x}_i)\|_1},$$

where the second to last step is a consequence of Dropout with probability 0.5 sampling for each layer uniformly at random from the set of all possible neuron activation patterns that can obtained by disabling some neurons of $\boldsymbol{s}_{t,l}(\boldsymbol{x}_i)$.

Term $\sum_{i=1}^{k} \sum_{l=d-1}^{1} \|\boldsymbol{s}_{t,l}(\boldsymbol{x}_i)\|_1$ can be seen as the sum of $k$ independent samples, each having mean $m = \text{avg}_{\boldsymbol{x} \in X} \sum_{l=d-1}^{1} \|\boldsymbol{s}_{t,l}(\boldsymbol{x})\|_1$ and maximum value $c = \max_{\boldsymbol{x} \in X} \sum_{l=d-1}^{1} \|\boldsymbol{s}_{t,l}(\boldsymbol{x})\|_1$. Hoeffding's inequality yields

$$P\left(\sum_{i=1}^{k} \sum_{l=d-1}^{1} \|\boldsymbol{s}_{t,l}(\boldsymbol{x}_i)\|_1 > E\left[\sum_{i=1}^{k} \sum_{l=d-1}^{1} \|\boldsymbol{s}_{t,l}(\boldsymbol{x}_i)\|_1\right] + k\delta\right) < \exp\left(-\frac{2\,k^2\delta^2}{kc^2}\right),$$

implying also that $P\left(2^{-\sum_{i=1}^{k} \sum_{l=d-1}^{1} \|\boldsymbol{s}_{t,l}(\boldsymbol{x}_i)\|_1} < 2^{-(k\mu + \sqrt{k/2}c)}\right) < 1/e$. Thus,

$$P\left(\tilde{\boldsymbol{S}} \text{ is a basis} \mid \boldsymbol{S} \text{ is a covering set}\right)$$

$$= P\left(\tilde{\boldsymbol{S}} \text{ is a basis} \mid \boldsymbol{S} \text{ is a covering set}, 2^{-\sum_{i=1}^{k} \sum_{l=d-1}^{1} \|\boldsymbol{s}_{t,l}(\boldsymbol{x}_i)\|_1} \geq 2^{-h}\right) P\left(2^{-\sum_{i=1}^{k} \sum_{l=d-1}^{1} \|\boldsymbol{s}_{t,l}(\boldsymbol{x}_i)\|_1} < 2^{-h}\right)$$

$$+ P\left(\tilde{S} \text{ is a basis} \mid S \text{ is a covering set}, 2^{-\sum_{i=1}^{k}\sum_{l=d-1}^{1}\|s_{t,l}(x_i)\|_1} < 2^{-h}\right) P\left(2^{-\sum_{i=1}^{k}\sum_{l=d-1}^{1}\|s_{t,l}(x_i)\|_1} < 2^{-h}\right)$$

$$\geq P\left(\tilde{S} \text{ is a basis} \mid S \text{ is a covering set}, 2^{-\sum_{i=1}^{k}\sum_{l=d-1}^{1}\|s_{t,l}(x_i)\|_1} \geq 2^{-h}\right) P\left(2^{-\sum_{i=1}^{k}\sum_{l=d-1}^{1}\|s_{t,l}(x_i)\|_1} < 2^{-h}\right)$$

$$\geq 2^{-(k\mu + c\sqrt{k/2})}\left(1 - 1/e\right) > 2^{-(k\mu + c\sqrt{k/2}+1)},$$

where the first step employs the law of total probability. We therefore deduce that

$$p_{\text{basis}}(S_T) \geq 1 - \left(\frac{p_{\text{cover}}(S)}{2^{(k\mu + c\sqrt{k/2}+1)}}\right)^{\lfloor \frac{|T|}{k} \rfloor}$$

$$= 1 - 2^{\left(\lfloor \frac{|T|}{k} \rfloor \log_2\left(\frac{p_{\text{cover}}(S)}{2^{(k\mu + c\sqrt{k/2}+1)}}\right)\right)}$$

$$= 1 - 2^{-\left(\frac{\lfloor \frac{|T|}{k} \rfloor}{(k\mu + c\sqrt{k/2}+1)}\log_2(1/p_{\text{cover}}(S))\right)},$$

which is satisfied with high probability when

$$|T| = \Omega\left(\frac{k(k\mu + c\sqrt{k/2}+1)}{\log\left(1/p_{\text{cover}}(S)\right)}\right) = \Omega\left(\frac{k^2\mu + nk^{3/2}}{-\log p_{\text{cover}}(S)}\right).$$

The final step of the proof entails bounding $\mu$ and $p_{\text{cover}}(S)$. We will think of neuron $i$ at layer $l$ as a (dependent) Bernoulli random variable with activation probability $p_l(i)$. The probability that neuron $i$ in layer $l$ is not activated within $k$ independent trials is $(1 - p_l(i))^k$. Taking a union bound over all neurons in all layers, results in:

$$p_{\text{cover}}(S) \geq 1 - \sum_{l=1}^{d-1}\sum_{i=1}^{n_{l-1}}(1 - p_l(i))^k = 1 - \sum_{l=1}^{d-1}\sum_{i=1}^{n_l}\left(1 - \frac{k\,p_l(i)}{k}\right)^k$$

$$\geq 1 - \sum_{l,i}\exp\left(-k\,p_l(i)\right)$$

$$\geq 1 - \exp\left(-kp_{min} + \log\left(\sum_{l=1}^{d-1}n_l\right)\right)$$

with $p_{min} = \min_{l,i} p_l(i)$. On the other hand, the average norm is given by

$$m = \operatorname*{avg}_{x \in X}\sum_{l=1}^{d-1}\|s_{t,l}(x)\|_1 = \sum_{x \in X}\frac{\sum_{l=1}^{d-1}\sum_{i=1}^{n_l}[s_{t,l}(x)](i)}{N}$$

$$= \sum_{l=1}^{d-1}\sum_{i=1}^{n_l}\frac{\sum_{x \in X}[s_{t,l}(x)](i)}{N}$$

$$= \sum_{l=1}^{d-1}\sum_{i=1}^{n_l}p_l(i) = \left(\sum_{l=1}^{d-1}n_l\right)p_{avg}.$$

The number of iterations we thus need to obtain a high probability bound is thus

$$|T| = \Omega\left(\frac{k^2\left(\sum_{l=1}^{d-1}n_l\right)p_{avg} + \left(\sum_{l=1}^{d-1}n_l\right)k^{3/2}}{-\log\left(1 - \exp\left(-kp_{min} + \log\left(\sum_{l=1}^{d-1}n_l\right)\right)\right)}\right).$$

If we select $k = (1 + o(1))\log\left(\sum_{l=1}^{d-1}n_l\right)/p_{min}$, we obtain

$$|T| = \Omega\left(\left(\frac{(1 + o(1))\log\left(\sum_{l=1}^{d-1}n_l\right)}{p_{min}}\right)^2\left(\sum_{l=1}^{d-1}n_l\right)p_{avg} + \left(\sum_{l=1}^{d-1}n_l\right)\left(\frac{(1 + o(1))\log\left(\sum_{l=1}^{d-1}n_l\right)}{p_{min}}\right)^{3/2}\right)$$

$$= \tilde{\Omega}\left(\left(\frac{1}{p_{min}}\right)^2 \left(\sum_{l=1}^{d-1} n_l\right) p_{avg} + \left(\sum_{l=1}^{d-1} n_l\right)\left(\frac{1}{p_{min}}\right)^{3/2}\right) = \tilde{\Omega}\left(\left(\sum_{l=1}^{d-1} n_l\right) \frac{p_{avg}}{p_{min}^2}\right),$$

where the asymptotic notation hides logarithmic factors.

The final Lipschitz constant is obtained by plugging in the bound of Observation 1 the value $k = (1 + o(1)) \log\left(\sum_{l=1}^{d-1} n_l\right)/p_{min}$.

$\square$

### B.7.2 Generalization

We prove the following:

**Lemma 4.** *In the setting of Lemma 3, suppose that the NN $f^{(t)}$ has been trained using a BCE loss and a sigmoid activation in the last layer, let $g^{(t)}(\boldsymbol{x}) = \mathbf{1}\left[f^{(t)}(\boldsymbol{x})) > 0.5\right] \in \{0, 1\}$ the classifier's output, and define*

$$r_t(X) := \frac{\min_{i=1}^N |f^{(t)}(\boldsymbol{x}_i) - 0.5|}{2\lambda_{f^{(t)}}^{bound}},$$

*where $\lambda_{f^{(t)}} \leq \lambda_{f^{(t)}}^{bound}$ with probability at least $1 - o(1)$. For any $\delta > 0$, with probability at least $1 - \delta - o(1)$, we have*

$$\left| E_{(\boldsymbol{x},y)}\left[er\left(g^{(t)}(\boldsymbol{x}), y\right)\right] - \underset{i=1}{\overset{N}{\mathrm{avg}}}\, er\left(g^{(t)}(\boldsymbol{x}_i), y_i\right)\right| \leq \sqrt{\frac{4\log(2)\,\mathcal{N}(\mathcal{X}; \ell_2, r_t(X)) + 2\log(1/\delta)}{N}},$$

*where $er(\hat{y}, y) = \mathbf{1}[\hat{y} \neq y]$ is the classification error and $\mathcal{N}(\mathcal{X}; \ell_2, r)$ is the minimal number of $\ell_2$-balls of radius $r$ needed to cover the input domain $\mathcal{X}$.*

*Proof.* For convenience, we drop the iteration index.

Following Xu and Mannor [2], we define the input margin $\gamma_i$ of classifier $g$ at $\boldsymbol{x}_i$ to be

$$\gamma_i := \sup\{a : \forall \boldsymbol{x},\ \|\boldsymbol{x} - \boldsymbol{x}_i\|_2 \leq a,\ g(\boldsymbol{x}) = g(\boldsymbol{x}_i)\},$$

which is the distance (in input space) to the classification boundary. For completeness, we also repeat the definition of a robust classifier:

**Definition 1** (Adapted from Definition 2 [2]). *Classifier $g$ is $(K, \epsilon)$-robust if $\mathcal{X} \times \mathcal{Y}$ can be partitioned into $K$ disjoint sets, denoted as $\mathcal{C}_{k=1}^K$, such that $\forall i = 1 \cdots, N$,*

$$(\boldsymbol{x}_i, y_i), (\boldsymbol{x}, y) \in \mathcal{C}_k \implies |er(g(\boldsymbol{x}_i), y_i) - er(g(\boldsymbol{x}), y)| \leq \epsilon.$$

Denote by $\boldsymbol{x}_i^*$ a point with $\|\boldsymbol{x}_i^* - \boldsymbol{x}_i\|_2 = \gamma_i$ with $g(\boldsymbol{x}_i^*) = g(\boldsymbol{x}_i)$ and notice that $f(\boldsymbol{x}_i^*) = 0.5$ (due to the definition $g(\boldsymbol{x}) = \mathbf{1}[f(\boldsymbol{x}) > 0.5]$). We use the argument of Sokolić et al. [3] and bound the input margin as follows:

$$\gamma_i \geq \frac{\|f(\boldsymbol{x}_i) - f(\boldsymbol{x}_i^*)\|_2}{\lambda_f} = \frac{\|f(\boldsymbol{x}_i) - 0.5\|_2}{\lambda_f} \geq \frac{\|f(\boldsymbol{x}_i) - 0.5\|_2}{\lambda_f^{bound}}, \tag{11}$$

with probability at least $1 - o(1)$. From Example 1 in [2] we then deduce that $g$ is $(2\mathcal{N}(2\mathcal{X}, \ell_2, r_t(X)), 0)$-robust for

$$r_t(X) = \frac{\|f(\boldsymbol{x}_i) - 0.5\|_2}{2\lambda_f^{bound}} \leq \min_{i=1}^N \frac{\gamma_i}{2}.$$

Theorem 3 [2] implies that if $g$ is $(K, 0)$-robust then, for any $\delta > 0$, the following holds:

$$\left| \mathrm{E}_{(\boldsymbol{x},y)}\left[er(g(\boldsymbol{x}), y)\right] - \underset{i=1}{\overset{N}{\mathrm{avg}}}\, er(g(\boldsymbol{x}_i), y_i)\right| \leq \sqrt{\frac{2\log(2)\,K + 2\log(1/\delta)}{N}}, \tag{12}$$

with probability at least $1 - \delta$. We obtain the final bound by substituting $K = 2\mathcal{N}(2\mathcal{X}, \ell_2, r_t(X))$ and taking a union bound on the events that inequalities (11) and (12) do not occur. $\square$

# C Additional theoretical results

## C.1 Generalization of Lemma 1 to any element-wise activation function

**Lemma 5.** *Let $f^{(t)}$ be a $d$-layer NN with arbitrary activation functions at the $t$-th SGD iteration, demote by $\boldsymbol{x}^{(t)} \in X$ the point of the training set sampled at that iteration, and set*

$$\epsilon_{f^{(t)}}(\boldsymbol{x}, y) := \left| \frac{\partial \ell(o, y)}{\partial o} \right|_{o = f^{(t)}(\boldsymbol{x})}. \tag{13}$$

*The Lipschitz constant of $f^{(t)}$ at $\boldsymbol{x}^{(t)}$ is*

$$\frac{\|\boldsymbol{b}_1^{(t+1)} - \boldsymbol{b}_1^{(t)}\|_2}{\alpha_t \cdot \epsilon_{f^{(t)}}(\boldsymbol{x}^{(t)}, y^{(t)})} \cdot \sigma_n(\boldsymbol{W}_1^{(t)}) \le \lambda_{f^{(t)}}(\boldsymbol{x}^{(t)}) \le \frac{\|\boldsymbol{b}_1^{(t+1)} - \boldsymbol{b}_1^{(t)}\|_2}{\alpha_t \cdot \epsilon_{f^{(t)}}(\boldsymbol{x}^{(t)}, y^{(t)})} \cdot \sigma_1(\boldsymbol{W}_1^{(t)}),$$

*where $\sigma_1(\boldsymbol{W}_1^{(t)}) \ge \cdots \ge \sigma_n(\boldsymbol{W}_1^{(t)}) > 0$ are the singular values of $\boldsymbol{W}_1^{(t)}$.*

*Proof.* The proof proceeds almost identically with that of Lemma 1. The main difference is that the diagonal matrix $\boldsymbol{S}_l^{(t)}(\boldsymbol{x}^{(t)})$ is redefined to yield the appropriate derivative for the activation function in question. Further, since now the function is not piece-wise linear, the bound only holds for $\boldsymbol{x}^{(t)}$ (and not for the entire region $\mathcal{R}_{\boldsymbol{x}^{(t)}}$ enclosing the point, as before). □

## C.2 The Lipschitz constant of the first layer

The behavior of SGD can also be indicative of the Lipschitz constant of the first layer when the training data is sufficiently diverse and the training has converged:

**Lemma 6.** *Let $f^{(t)}$ be a $d$-layer NN trained by SGD, let Assumption 1 hold, and further and suppose that after iteration $\tau$, we have*

$$\frac{\|\boldsymbol{W}_2^{(t+1)} - \boldsymbol{W}_2^{(t)}\|_2}{\|\boldsymbol{b}_1^{(t+1)} - \boldsymbol{b}_1^{(t)}\|_2} + \|\boldsymbol{b}_1^{(t)}\|_2 \le \vartheta \quad and \quad \|\boldsymbol{W}_1^{(t)} - \boldsymbol{W}_1^{(t')}\|_2 \le \beta \quad for\ all \quad t, t' \ge \tau.$$

*Denote by $\delta$ the minimal scalar such that, for every $\boldsymbol{x} \in \mathcal{X}$, we have $\|\boldsymbol{x} - \boldsymbol{x}_i\|_2 \le \delta$ for some $\boldsymbol{x}_i \in X$. Then,*

$$\lambda_{f_1^{(t)}} \le \frac{\vartheta + \beta}{1 - \delta}. \tag{14}$$

*under the condition $\delta < 1$.*

*Proof.* The weight matrix gradient is at a point $\boldsymbol{x}$ is

$$\left( \frac{\partial f(\boldsymbol{x}, \boldsymbol{w}^{(t)})}{\partial \boldsymbol{W}_l^{(t)}} \right)^\top = f_{l-1}^{(t)}(\boldsymbol{x}, \boldsymbol{w}^{(t)}) \cdot \boldsymbol{W}_d^{(t)} \cdots \boldsymbol{S}_{l+1}^{(t)}(\boldsymbol{x}) \boldsymbol{W}_{l+1}^{(t)} \boldsymbol{S}_l^{(t)}(\boldsymbol{x}).$$

Fixing

$$\left\| \left( \frac{\partial \ell(f(\boldsymbol{x}^{(t)}, \boldsymbol{w}^{(t)}), y^{(t)})}{\partial \boldsymbol{W}_l^{(t)}} \right)^\top \right\|_2 \left\| \left( \frac{\partial \ell(f(\boldsymbol{x}^{(t)}, \boldsymbol{w}^{(t)}), \boldsymbol{y}^{(t)})}{\partial \boldsymbol{b}_{l-1}^{(t)}} \right)^\top \right\|_2^{-1} \le \varrho_l(\boldsymbol{x}^{(t)})$$

we have that

$$\left\| \left( \frac{\partial \ell(f(\boldsymbol{x}^{(t)}, \boldsymbol{w}^{(t)}), y^{(t)})}{\partial \boldsymbol{W}_l^{(t)}} \right)^\top \right\| = \left\| f_{l-1}(\boldsymbol{x}^{(t)}, \boldsymbol{w}^{(t)}) \right\| \left\| \frac{\partial \ell(\hat{y}, y)}{\partial \hat{y}} \boldsymbol{W}_d^{(t)} \boldsymbol{S}_{d-1}^{(t)}(\boldsymbol{x}^{(t)}) \cdots \boldsymbol{W}_{l+1}^{(t)} \boldsymbol{S}_l^{(t)}(\boldsymbol{x}^{(t)}) \right\|$$

$$= \left\| f_{l-1}(\boldsymbol{x}^{(t)}, \boldsymbol{w}^{(t)}) \right\|_2 \left\| \left( \frac{\partial \ell(f(\boldsymbol{x}^{(t)}, \boldsymbol{w}^{(t)}), y^{(t)})}{\partial \boldsymbol{b}_{l-1}^{(t)}} \right)^\top \right\|_2,$$

which implies

$$\left\| f_{l-1}(\boldsymbol{x}^{(t)}, \boldsymbol{w}^{(t)}) \right\| \leq \varrho_l(\boldsymbol{x}^{(t)}). \tag{15}$$

Let $\boldsymbol{x}^* = \mathrm{argmax}_{\boldsymbol{x} \in \mathcal{S}_{n-1}} \left\| \boldsymbol{S}_1^{(t)}(\boldsymbol{x}) \boldsymbol{W}_1^{(t)} \boldsymbol{x} \right\|_2$ and fix $\boldsymbol{x}^{(t')}$ to be the point in the training set that is closest to it (sampled at iteration $t' \geq \tau$).

$$\lambda_{f_1^{(t)}} = \left\| \boldsymbol{S}_1^{(t)}(\boldsymbol{x}^*) \boldsymbol{W}_1^{(t)} \boldsymbol{x}^* \right\|_2 \leq \left\| \boldsymbol{S}_1^{(t)}(\boldsymbol{x}^{(t')}) \boldsymbol{W}_1^{(t)} \boldsymbol{x}^{(t')} \right\|_2 + \left\| \boldsymbol{S}_1^{(t)}(\boldsymbol{x}^*) \boldsymbol{W}_1^{(t)} \boldsymbol{x}^* - \boldsymbol{S}_1^{(t)}(\boldsymbol{x}^{(t')}) \boldsymbol{W}_1^{(t)} \boldsymbol{x}^{(t')}) \right\|_2.$$

By the main assumption, we can bound the rightmost term by $\| \boldsymbol{x}^* - \boldsymbol{x}^{(t')} \| \lambda_{f_1^{(t)}} \leq \delta \lambda_{f_1^{(t)}}$. We thus get

$$
\begin{aligned}
\lambda_{f_1^{(t)}} = \left\| \boldsymbol{S}_1^{(t)}(\boldsymbol{x}^*) \boldsymbol{W}_1^{(t)} \right\|_2 &\leq \left\| \boldsymbol{S}_1^{(t)}(\boldsymbol{x}^{(t')}) \boldsymbol{W}_1^{(t)} \boldsymbol{x}^{(t')} \right\|_2 + \delta \lambda_{f_1^{(t)}} \\
&\leq \left\| \boldsymbol{S}_1^{(t)}(\boldsymbol{x}^{(t')}) \boldsymbol{W}_1^{(t')} \boldsymbol{x}^{(t')} \right\|_2 + \left\| \boldsymbol{W}_1^{(t')} - \boldsymbol{W}_1^{(t)} \right\|_2 + \delta \lambda_{f_1^{(t)}} \\
&\leq \left\| \boldsymbol{S}_1^{(t')}(\boldsymbol{x}^{(t')}) \boldsymbol{W}_1^{(t)} \boldsymbol{x}^{(t')} + \boldsymbol{b}_1^{(t')} \right\|_2 + \| \boldsymbol{b}_1^{(t')} \|_2 + \left\| \boldsymbol{W}_1^{(t')} - \boldsymbol{W}_1^{(t)} \right\|_2 + \delta \lambda_{f_1^{(t)}} \\
&= \varrho_2(\boldsymbol{x}^{(t')}) + \| \boldsymbol{b}_1^{(t')} \|_2 + \left\| \boldsymbol{W}_1^{(t')} - \boldsymbol{W}_1^{(t)} \right\|_2 + \delta \lambda_{f_1^{(t)}} \\
&\leq \vartheta + \left\| \boldsymbol{W}_1^{(t')} - \boldsymbol{W}_1^{(t)} \right\|_2 + \delta \lambda_{f_1^{(t)}},
\end{aligned}
$$

where, due to Assumption 1, $\boldsymbol{S}_1^{(t')}(\boldsymbol{x}^{(t')}) = \boldsymbol{S}_1^{(t)}(\boldsymbol{x}^{(t')})$. The final bound is obtained re-arrangement and by the convergence assumption $\left\| \boldsymbol{W}_1^{(t')} - \boldsymbol{W}_1^{(t)} \right\|_2 \leq \beta$. $\qquad\square$