# OpenReview forum: "What training reveals about neural network complexity"
_NeurIPS.cc/2021/Conference — NeurIPS 2021 Poster_

### Official Review · Reviewer_7TvM · 2021-07-14

**Rating:** 6
**Confidence:** 3

**Summary:**

The authors measure the complexity of the learned function of NN by the distribution of an approximation of its Lipschitz constants. They analytically show that the average of the Lipschitz constants at the beginning of the learning process can predict various aspects of the entire learning process, which could possibly allow picking the right architecture in a faster way.

**Limitations And Societal Impact:**

not relevant

**Main Review:**

The paper tackles an interesting topic, from a theoretical approach. As far as I can tell, this approach is new and can shed light on several aspects of NN training dynamics, which could be beneficial to the community.

However, this paper also has many faults:
The paper is very hard to read and even harder to understand. It is composed of many equations, with a stunning amount of notations and abbreviations. The paper introduces a large number of bounds, and the distinctions between them could be better explained. The paper lacks some high-level overview, which can better explain the motivation and usage of each bound.

The empirical part of the work feels premature. The authors merely explain the settings of each experiment, without clear motivation or analysis, which again makes it very hard to follow. The experiments are done mostly in toy settings, and it is unclear if they could be extended to real problems, raising concerns about the validity of the motivation.


Since this work is not directly in my field of studies, some of my concerns may be due to my unfamiliarity. Therefore, I tend to accept this paper, but as I can not give it a higher score than "marginally accept" with its current issues.


----------------------------------------------------------

Post rebuttal:
After reading the authors' response and the other reviews, I'm more confident than my original review that this is indeed a good paper, which should be accepted. As other reviewers also mentioned the readability issues, which in my opinion, are a significant problem with this version, I'll keep my score as "6: Marginally above the acceptance threshold", and raise my confidence score to 3.

**Time Spent Reviewing:**

4

---

> ### Author Response · Authors · 2021-08-10
> **Reply to reviewer 7TvM**
>
> Thank you for your effort in doing this review. We reply to your criticisms below:
>
> ### 1. Complexity of equations
>
> We understand the reviewer's concern that the mathematical findings of the paper are complicated.
>
> We commit to doing the following in the camera-ready version:
>
> * Considerably simplify the statements of all technical results in Section 4. To do this, we will express things in terms of the per-point Lipschitz constant (rather than the average over the training set). This will simplify the technical results significantly, allowing us to eliminate many definitions and remove the complex expressions involving the length of the time interval T.
>
> * Enhance our intuitive explanations of the theoretical bounds. This includes discussing why we focus on the bias (lines 150-156), what the condition of Theorem 2 means (lines 231-234), explaining Assumption 1, and providing intuition for the role of Dropout in Theorem 3.
>
> ### 2. Role of experiments
>
> We believe that the reviewer has misunderstood the role of our experimental section.
>
> Our experiments do not intend to explore the relation of training and complexity/generalization. These relations have been explored in previous large-scale experimental studies (papers [1] and [3] cited in the introduction) giving a solid empirical motivation to the questions that we study in this work.
>
> Our paper's contribution is to render *mathematically concrete* observations about some relations between training and complexity. For this reason, our numerical results target specifically our theoretical results and aim to validate our theory.
>
> * Figure 1 tests the prediction of Theorem 1 that bias update speed correlates with NN complexity in two tasks (toy regression, CIFAR) and with two architectures (MLP and CNN).
>
> * Figure 2 tests the predictions of Corollaries 2 and 3 about the distance to initialization and variance of $b_1$.
>
> * Figure B.3 explores the predictions of Theorem 2 about the Lipschitz constants of empty regions.
>
> * The appendix includes further ablations, testing the effect of the architecture on MNIST, and showing the effect of batching.
> Though more large-scale experiments are always possible, we believe that these numerical studies do well to support our theoretical claims.
>
> To avoid future misunderstandings, we will update the introductory paragraph of Section 6 to explain that our experiments simplify aim to validate our theoretical results and that readers that are interested in a more general empirical study can refer to [1] and [3].

---

### Official Review · Reviewer_j63f · 2021-07-15

**Rating:** 7
**Confidence:** 3

**Summary:**

The paper presents an analysis of learning dynamics of neural network that relate the trajectory of changes in the first layer bias of neural network to bounds on Lipschitz constant, and thus the complexity of the model.   In other words, the authors establish that behaviour of first layer’s bias during training is a good predictor of the trained model’s complexity.

**Main Review:**

Though the main logic of the argument is somewhat straight forward – Lipschitz constant of a network is linked to complexity, gradient of the entire network is linked to Lipchitz constant, therefore the trajectory of first layer bias, which depends on that gradient, is linked to complexity of the entire model – I think a rigorous mathematical treatment and the results presented are worth of notice.  The paper is written well and the ideas are presented clearly.

I think the results in Ch 4 for complexity around the training points are interesting, though it is clear that from practical point of view, trajectory of bias alone is not sufficient to gauge complexity, as one also has to take into account singular values of W_1, which would change during training.

I do have a bit of an issue when it comes to complexity analysis when “far from training data”.  Specifically, if my understanding of Assumption 1 in Ch 5 is correct,  it allows changes of parameters in the model as long as long as the activation patterns (which ReLU neurons switch on and which one stay off)  stay the same.  Doesn’t that make the neural network essentially a linear model?  While this makes the analysis possible, it throws away the thing that makes neural networks useful…and I suspect, the Assumption is not likely to hold in practice


**Time Spent Reviewing:**

5

---

> ### Author Response · Authors · 2021-08-10
> **Reply to reviewer j63f**
>
> Thank you for your review and comments.
>
> ### 1. Results of Section 4
>
> We agree with you that, though important, the trajectory of bias is not telling the whole story. The largest singular value of the first layer is one aspect when upper bounding the NN complexity (though, usually, its value is a modest constant so that should not have a huge effect on the bounds). Another aspect is the derivative of the loss $\epsilon_t$ which serves as a normalization of the bias. So yes, to bound the Lipschitz constant close to training data one needs to look at all three quantities (bias update, singular values, and $\epsilon_t$). Our experiments also support this claim by showing that the correctly-normalized bias update does correlate with the NN complexity and its ability to generalize.
>
> ### 2. Assumption 1 and complexity analysis far from training data
>
> Thank you for your question. We believe that you have misunderstood Assumption 1: we assert that the activation pattern of each point $x$ stays the same across $t\geq \tau$ (not that all points have the same activation pattern). Therefore, the NN can still have different activation patterns at different points, as long as these activations stay persistent over successive iterations. In effect, we are assuming that the way the NN partitions the input space remains the same after $\tau$. For instance, our assumption will be practically satisfied at the end of the training if one decreases the learning rate to a very small value.
>
> Thus, the technical results concern highly non-linear functions and are non-trivial. We hope that this explanation clarifies things.
>
> We will update the paper to make this point explicit.

---

> > ### Comment · Reviewer_j63f · 2021-09-02
> > **Re: Response**
> >
> > Thanks for answering my questions. And although my reservations about Assumption 1 do not completely go away (because despite it being less restrictive than I originally thought, I am still not sure whether there is any chance it would hold in practice), I don't see it as a major problem for the purpose of theoretical analysis.  And your response on further clarification of experiment B.5 was satisfactory to me, and so I have raised my score.

---

### Official Review · Reviewer_i77r · 2021-07-16

**Rating:** 7
**Confidence:** 4

**Summary:**

The paper attempts to understand the complexity of the function learned by a model from the training behavior of its weights. The authors consider neural networks with ReLU activation. The 3 main results are:

(a) The length and variance of the trajectory of the bias of the first layer can be bounded by a function of the Lipschitz constant close to training data.

(b) The Lipschitz constant (for steady learners) can also be given for regions not present in the training data, provided the activation pattern in the region is covered by the activation patterns of the training data.

(c) The authors give a data-dependent and training-dependent generalization bound for steady learners with Dropout. The proof involves bounding the global Lipschitz constant, which requires a simplified assumption that all the neurons have non-zero activity without Dropout in the training set.

The authors corroborate their findings with an experimental study.

**Limitations And Societal Impact:**

The authors mention the limitations and the future directions of their work. Since the paper discusses a theoretical connection between trajectory and the Lipschitz constant of the neural network, I don't foresee a potential negative impact (to the best of my knowledge).

**Main Review:**

The paper tries to understand the relationship between the SGD trajectory and the complexity of the learned function. The authors come up with a novel strategy to look at the trajectory of the bias of the first layer. The novelty of the technique comes from cleverly utilizing the fact that a ReLU network divides the input space into multiple linear regions and then connecting the bias to the Lipschitz constant of the linear classifier in each region. A couple of previous works try to connect the trajectory of weights of the model to the complexity of the function learned. However, most of the analyses were in the NTK or the mean-field regime. The strength of this work is that it assumes nothing on the initialization of the neural network.

However, I have a couple of questions and few suggestions:

(1) In theorem 1, the assumption to keep the upper bound finite is $\sigma_n (W_1^{(t)}) > 0$. I think the authors should explicitly mention it in the theorem statement. What happens when the data comes from a low dimensional space $k < n$, and the input layer weights have to fit that subspace? Can the bounds then be changed to $\sigma_k(W_1^{(t)})$ in some trivial way?

(2) What do the authors mean by the statement "a larger complexity NN will need to fit the training data more closely to achieve the same variance as that of a lower complexity NN" (line 189)? What parameters in Corollary 2 do I need to compare to get this statement (e.g. $\epsilon^{\mathrm{var}}_{ f^{(t)} }$, and the Lipschitz constant)?

 (3) I believe you need more conditions to claim Corollary 1. I think the result that you can claim from theorem 1 and definition 1 is that given a set of iteration indices $T \ge \tau$ for a $(\tau, \varphi)$-steady learner, with $|T|$ large enough, $\frac{1}{T} \sum_{t \in T} \lambda^{avg}_{f^{(t)}} (X) \le \beta \varphi$.  How is this related to the Lipschitz constant of every training data point after $t \ge \tau$, provided you don't assume how the Lipschitz constant changes for a steady learner?

(4) Can the authors explain how Dropout helps in Theorem 3? I guess, Claim 1 in the appendix is the main underlying tool. However, the proof contains multiple notational issues (e.g. what is the meaning of $\sum_{i=1}^{k} s_{ t, l }(x)$?). But I can guess the strategy is that since Dropout drops neurons with a probability of $1/2$,  and the assumption that all the neurons have some activity in the training set, one can show that the activation patterns of the training set (along with Dropout masks) can form a basis of the activation pattern of the entire region with some non-zero probability. That will now help to get a bound on the global Lipschitz constant from Theorem 2.

It will be great if the authors can mention 2-3 lines on how dropout helps in the main paper.

(5) In the experiments on regression of a sinusoidal function, the authors have set the first layer of the MLP to Identity. How do the experimental observations change if they don't fix it to Identity?

(6) It will be interesting to see if the experimental observations stay the same for trajectories of biases of the other layers.

**Notation issues

$\lambda_f^{\mathrm{steady}}$ has never been defined in the main paper.

I enjoyed reading the paper since it makes a novel attempt at understanding the connection between the trajectory of neural networks and the complexity of the function learned. The proofs are simple to understand (baring notational issues in the appendix), which is a strength of this paper. My scores are slightly lower since I need to make sure I understand the use of Dropout in theorem 3. I am happy to increase my score after discussing it with the authors and other reviewers.


***After Rebuttal***
I have increased my score to 7. I would request the authors to make changes to the paper to address the different clarity issues mentioned by the reviewers.

**Time Spent Reviewing:**

5 - 7 hours

---

> ### Author Response · Authors · 2021-08-10
> **Reply to reviewer i77r**
>
> Thank you for your thorough and constructive feedback!
> We believe that your review has helped us to improve the quality of our paper.
>
> We answer your comments below:
>
> ### 1. Singular values in Theorem 1
> Good point. We will update the theorem statement to explicit that $\sigma_n(W_1^{(t)})>0$.
>
> The reviewer also asked if the condition on the minimal singular value can be easily relaxed when the data fall in some lower-dimensional space. Indeed, if all $x_i$ fall within some $k$-dimensional subspace $S_k$ then, rather than the minimum singular value, we can update the upper bound numerator to depend on $\min_{x \in S_k } || W_1^{(t)} x ||_2$ rather than $\sigma_n(W_1^{(t)})>0$. We will try to fit an explanation of this in the camera-ready version.
>
> ### 2. Statement about Corollary 2 (line 189)
>
> The reviewer asked what do we mean by "*a larger complexity NN will need to fit the training data more closely to achieve the same variance as that of a lower complexity NN*".
>
> The statement refers to the role of $c$ in the bound.
>
> * To simplify things let's say that $\epsilon^{var}_{f^{(t)}} = 1$ and that $\delta = o(1)$.
>
> * Then, $c \approx 1 / avg_{t \in T} (\epsilon^{avg}_{f^{(t)}})^4$.
>
> * It follows that the variance grows with $avg_{t \in T} (\epsilon^{avg}_{f^{(t)}})^4$, which is smaller when the NN fits the training data better.
>
> Thus, the variance bound can decrease either by (a) considering a NN with a smaller Lipschitz constant close to the training data or (b) by keeping the NN's complexity the same but fitting the data more closely.
>
> To render the point above more clear, we will simplify the statement of the corollary (particularly of the constant $c$). We believe that putting the $\epsilon$ terms within the main expression of the bound will make the relations between the terms easier to grasp.
>
> ### 3. Corollary 1
>
> We agree that the statement of Corollary 1 can be made more precise. We will update the statement to write that, if we select an iteration $t> \tau$ at random from the interval $T$, the Lipschitz constant of $f^{(t)}$ at every data point $x_i$ will be bounded by $\beta \varphi$ generically (i.e., with probability that converges to 1 as $|T|$ grows).
>
> We include here a proof sketch for reference (we will also add the full proof in the supplementary):
>
> **PROOF (sketch).**
> For any point $x$ with label $y$ and iteration $t$, we say that condition $c_t(x)$ holds if  $\|b_1^{(t+1)} - b_1^{(t+1)}\| \leq \varphi \alpha_t \epsilon_{f^{(t)}} (x, y)$ (this is the same condition as in the Corollary statement but applied to an arbitrary point $x$). Write $\kappa_t$ to refer to the number of training points $x_i \in X$ for which $c_t(x_i)$ holds. Clearly, $\kappa_t \in [0,N]$.
> Moreover, suppose that there are $\xi$ iterations within $T$ for which $\kappa_t <N$ (these are the bad iterations where the Lipschitz constant is larger than $\beta \varphi$ for at least one point in the training set).
> Since we sample $x^{(t)}$ i.i.d. from $X$, the probability that $c_t(x^{(t)})$ is satisfied for every $t \in T$ (this is the probability that the condition of the corollary is met) is at most $\prod_{t \in T} (\frac{\kappa_t}{N}) \leq ((N-1)/N)^\xi = (1 - 1/N)^\xi \leq e^{-\xi/N}$. Asymptotically, for the latter probability to be larger than 0, it must be that $\xi$ doesn't grow with $|T|$.
> Now, if we select the iteration $t$ at random from $T$, the probability that there will be some $x_i \in X$ for which $c_t(x_i)$ is not satisfied is $\xi/|T| = O(1/|T|)$, which converges to 0 as $|T|$ grows. **END**
>
> ### 4. How Dropout helps in Theorem 3
> Your intuition is correct: when we are dropping neurons (under the condition that all neurons are eventually active -- or equivalently that the activations form a covering set) eventually a set of activations forming a basis will appear. The derivation then bounds the probability that a valid basis will be seen as a function of the length of the interval.
>
> Thank you for spotting the notation issue: the term in question misses the index $i$ (i.e., it should be $\sum_{i=1}^k s_{t,l} (x_i)$). The same index is missing throughout the proof of Claim 1. We will fix this. As per your recommendation, we will also add a short explanation of the intuition behind why Dropout helps in the main paper (directly after line 243).
>
> ### 5 \& 6. Experiments
>
> * matrix $W_1$: We haven't tried this yet for computational efficiency (we set it to identity to avoid computing the singular value after every iteration.) We will include in the camera-ready version an experiment that tests this.
>
> *  biases $b_k$: We suspect that they will be more weakly correlated. A key issue is that for higher biases $b_k$ the bounds become more vacuous exponentially with $k$. This is because we would need to upper bound the Lipschitz constant of the subnetwork consisting of the first $k$ layers with a product of spectral norms bound.
>
> ### Other comments
>
> The definition of $\lambda_{f^{(t)}}^{steady}$ can be found in the equation after line 243 in the main paper.

---

> > ### Comment · Reviewer_i77r · 2021-08-24
> > **Discussion cont'd**
> >
> > Thank you for the detailed response. As a part of internal discussion, some of my fellow reviewers have concerns about the quality of experiments in section B.5. Is it possible to have some quick experiments as discussed below?
> >
> > Basically, the concerns of the reviewers are twofold:
> >
> > a) The reviewers think that the experiments in which just optimization hyper-parameters are varied (e.g. learning rate) would be a much better choice, compared to the MLP vs CNN experimental setting. This is because the theorems in the paper talk about the relation between NN complexity and bias trajectory for a __fixed class of NN function__. Therefore the reviewers are not sure if it makes sense to compare the bias trajectory length of an MLP to that of a CNN.
> >
> > b) As evident in Figure B.5 (d) CNN has a lower training loss (especially visible in the MNIST1000 case). That means MLP, if anything, is "underfitting" and not "overfitting"...which could suggest MLP delivers a less complex function that doesn't generalize because it is not complex enough. It seems vacuous to deduce from bias trajectory that CNN is less complex and use this as evidence that bias trajectory is correlated with generalization. The reviewers think the complexity of the function needs to be judged on the test performance of models trained to the same level of loss. Perhaps it would be possible to run the experiment for a bit longer to get both models to achieve identical train loss? Or maybe the complexity of the function is a red herring here. Maybe the CNN, despite being more complex, generalizes better because it simply has a better bias (in the sense of inclination/prejudice) for the target problem, and therefore its parameters (including bias weights) need less adjustment...and that's a good indicator of generalization regardless of function complexity.
> >
> > Looking forward to the authors' responses.

---

> > > ### Author Response · Authors · 2021-08-25
> > > **Regarding the experiment of Section B.5**
> > >
> > > Thank you for bringing these concerns to our attention. We discuss them below:
> > >
> > > > a) The reviewers think that the experiments in which just optimization hyper-parameters are varied (e.g. learning rate) would be a much better choice, compared to the MLP vs CNN experimental setting. This is because the theorems in the paper talk about the relation between NN complexity and bias trajectory for a __fixed class of NN function__. Therefore the reviewers are not sure if it makes sense to compare the bias trajectory length of an MLP to that of a CNN.
> > >
> > > Thank you for the suggestion.
> > >
> > > We would like to clarify that our *Theorems enable one to directly compare the Lipschitz constant of any neural network that is described by eq. in line 108 of our paper, including those with fully-connected and convolutional layers* (see also discussion in lines 166-171). Indeed, we can recall that a convolutional layer can be expressed as a fully-connected layer $\rho(Wx+b)$ where $W$ is constrained to have a specific structure (usually sparse and Toeplitz). We also remark that Lipschitz continuity (and thus the normalized bias trajectory) is a relevant measure of complexity for both MLP and CNN architectures alike (as it is a property of a function that is expressed w.r.t. the input space and not the weight space). For these reasons, we believe that a comparison between the two architectures is meaningful.
> > >
> > > We have considered the effect of optimization hyperparameters (specifically batch size) in our supplementary section B.6. We also experimented with the learning rate. We found that, since we are using pure SGD, altering the learning rate often significantly slowed down the convergence, with the NNs loss often not becoming small enough even after *very* long training periods. For this reason, we deemed that a more exhaustive investigation was impractical.
> > >
> > > > b) As evident in Figure B.5 (d) CNN has a lower training loss (especially visible in the MNIST1000 case). That means MLP, if anything, is "underfitting" and not "overfitting"...which could suggest MLP delivers a less complex function that doesn't generalize because it is not complex enough. It seems vacuous to deduce from bias trajectory that CNN is less complex and use this as evidence that bias trajectory is correlated with generalization. The reviewers think the complexity of the function needs to be judged on the test performance of models trained to the same level of loss. Perhaps it would be possible to run the experiment for a bit longer to get both models to achieve identical train loss? Or maybe the complexity of the function is a red herring here. Maybe the CNN, despite being more complex, generalizes better because it simply has a better bias (in the sense of inclination/prejudice) for the target problem, and therefore its parameters (including bias weights) need less adjustment...and that's a good indicator of generalization regardless of function complexity.
> > >
> > > Good points. We would like two make two remarks here:
> > >
> > > **1. Under vs overfitting**
> > >
> > > The MNIST task becomes quite easy given more than a few hundred training samples. The latter can be seen from that both MLP and CNN attain good test error and generalization in the MNIST1000 experiment. We included these subfigures because we believe that it's important to present the reader with a complete picture of the task.
> > >
> > > *Following your suggestion, we will run the MNIST1000 experiment longer to see what happens as the training loss of the MLP keeps decreasing.*
> > >
> > > Nevertheless, it is important to note that the classification task becomes slightly harder with fewer training samples (i.e., in the case of MNIST100). As it can be seen in Figure B.5 (a-c), at 20k iterations both MLP and CNN attain close to zero training loss, while the CNN generalizes better to unseen test examples. We are thus in the overfitting regime.
> > >
> > > **2. Complexity of function and inductive bias of CNN**
> > >
> > > We completely agree with the reviewer that the complexity of the learned function is not the sole factor driving generalization (though it is can be a crucial factor all other things being equal). Convolutional layers are indeed more constrained and better suited to image data than fully convolutional ones -- thus it is reasonable to expect better generalization than MLPs. We also fully agree that generalization needs to be judged on the test performance of models trained to the same level of loss.
> > >
> > > We included the experiment because we were curious to find out how the NN complexity (and thus the normalized bias trajectory length) would differ amongst two models whose inductive bias aligns or misaligns with the task. Our experiment shows that the CNN, beyond having the right architecture for the task, also learns a slightly lower complexity function than the MLP while fitting the training data equally well or better. Thus, here we mainly use the bias trajectory length as a diagnostic tool that helps us understand what functions the two architectures are learning. The argument of the reviewer about the parameters of the CNN needing less adjustment from initialization applies w.r.t. the total parameter trajectory length and not the normalized per-iteration one that we examine here: even if the bias needs less adjustment overall, as our Theorems show, the normalized rate of change will be indicative of the NN's Lipschitz constant.
> > >
> > > We will update the camera-ready document to explicit that (a) the Lipschitz constant is not the sole factor determining generalization and (b) that our claim that CNN generalizes better applies only to the MNIST100 case. As mentioned above, we will also run the MNIST1000 experiment longer.

---

### Official Review · Reviewer_AHNy · 2021-08-02

**Rating:** 7
**Confidence:** 3

**Summary:**

This paper investigates how certain properties of the training trajectory such as its length relate to the complexity of functions learned by ReLU networks. Complexity is quantified by way of the network's distribution of the Lipschitz constants on the linear regions of its input space. The key finding is that networks that fit the training data with a smaller average Lipschitz constant exhibit a shorter trajectory and their parameters vary less near training convergence. For training with dropout, the authors also give a data-dependent generalization bound that features a polylogarithmic dependence on the number of parameters under certain assumptions.


**Limitations And Societal Impact:**

Addressed by the authors. I do not foresee any potential negative societal impact of this work.


**Main Review:**

I found the paper reasonably well-written and the results presented coherently. The techniques used in the paper in relation to bounding the Lipschitz constant by exploiting certain properties of ReLU networks appear novel.

The authors quantify neural network complexity by way of the Lipschitz continuity of the learned function on the linear regions of the input space. For ReLU networks, locally the Lipschitz constant is given by the norm of the gradient at any point within that region. The authors build on the observation that if the learned function has a large Lipschitz constant near an input datum, then SGD updates the bias of the first layer more quickly. Thus, for linear regions close to the training data, one can infer the average Lipschitz constant of the learned function by observing how fast the bias is updated. The authors show that the length of the optimization trajectory (with high probability over the SGD sampling) grows linearly with the average Lipschitz constant of the network.

Simple experiments studying the training behavior on two tasks, namely, fitting sinusoids of increasing frequency and classifying CIFAR-10 images with increasing label corruption lend support to the theory: given two networks that fit the training data equally well, the one implementing a higher complexity function exhibits a longer trajectory with the weights having bigger variance near training convergence. The authors also show how the analysis can be extended to regions of the input space that are outside the training data.


The proof for the upper bound in Theorem 1 assumes that the smallest singular value $\sigma_n(W_1^{(t)})$ is positive. This condition is missing in the statement of Theorem 1.

Minor:

l.123 (Sentence): that yield same activation pattern --> that yield the same activation pattern

l.229 (Sentence) "Furthermore, denote by ... used in the last layer." -- add, respectively: Let $S_T$, $\varphi_T$, $\mu_T$ denote, respectively, ...






**Time Spent Reviewing:**

3.5

---

> ### Author Response · Authors · 2021-08-10
> **Reply to reviewer AHNy**
>
> Thank you for reading our work and for your positive and constructive comments.
>
> You are correct about the condition on the smallest singular value of $W_1$: when the singular value is zero the upper bound goes to infinity. We will add the corresponding condition in the statement of Theorem 1.
>
> We will also incorporate your other minor suggestions.

---

### Official Review · Reviewer_juSM · 2021-08-03

**Rating:** 6
**Confidence:** 3

**Summary:**

Note: this is a light review

The paper makes the hypothesis that the complexity of the function learned by DNNs depends on the convergence rate of its parameters (with special focus on the bias of the 1st layer) and other factors (variance of the bias, and distance from initialization).



**Limitations And Societal Impact:**

please see the main review

**Main Review:**

Clarity: The paper is not clearly written. For instance, the meaning of the term "steadily" is not clear in the abstract (it is defined formally later in the paper). The statements of theoretical results are not explained in text to provide an intuitive understanding.

Major problems:

Some of the claims made by the authors are not concrete (but are rather casual). For instance, the first line of introduction says "Though neural networks (NNs) trained on relatively small datasets can generalize well, often significant trial and error is needed to select an architecture that does not overfit.". This statement has 2 problems:

1. typically using a ResNet architecture works out-of-the-box for image datasets and yields SOTA performance on classification tasks.
2. the claim that a proper selection of architecture can result in a model avoiding overfitting all together in practice is simply not true.

There is a mismatch between the claims made in the abstract/introduction section and the experimental results. The claims are mainly talking about the connection between behavior of the training process and the complexity of the DNN function learned (line 21). This suggests that the authors are considering the case where given a fixed dataset, if we change optimization hyper-parameters (or other factors), it will result in a change in the behavior of the training process which should be indicative of the complexity of the learned DNN function. However, the experiments use datasets with different levels of corruption or target functions with varying complexity (as measured by their frequency components). These experiments do not shed light on whether the behavior of training process can be indicative of the learned hypothesis (because to study that, one would have to fix the target function). Another problem is that these experimental findings are not novel. It has been shown in [1] that the complexity of the target function impacts the speed of convergence of DNN training. The additional plot provided by authors involving the converge rate of the bias of the first layer of the DNN is not surprising because of this simple reasoning: if the training/test loss has not converged, the parameters of the DNN must not have converged also.

Further, I find the choice of making the bias of the 1st hidden layer of a DNN the main object of analysis (viz complexity of the learned function) debatable. Any useful information/structure about the data is captured by the DNN weights, not biases. Biases merely act as threshold values (when using ReLU activations).

[1] Arpit, D., Jastrzębski, S., Ballas, N., Krueger, D., Bengio, E., Kanwal, M. S., ... & Lacoste-Julien, S. (2017, July). A closer look at memorization in deep networks. In International Conference on Machine Learning (pp. 233-242). PMLR.

**Time Spent Reviewing:**

4-5 hours

---

> ### Author Response · Authors · 2021-08-10
> **Reply to reviewer juSM**
>
>
> Thank you for your comments.
>
> Let us start by stressing that the paper's main contribution is theoretical (i.e., proving a relationship between bias, Lipschitz constant, and generalization). Thus, we hope that its merit will be judged primarily based on these results.
>
> Below we discuss your main comments:
>
> ### 1. Clarity
>
> As per your suggestion, we will improve the clarity of the camera-ready version by (a) removing the word steadily from the abstract and (b) providing further intuitive explanations of the theoretical results (e.g., discussing why we focus on the bias (lines 150-156), what the condition of Theorem 2 means (lines 231-234), explaining Assumption 1, and providing intuition for the role of Dropout in Theorem 3).
>
> ### 2. The first line of the introduction
>
> The reviewer expressed concerns regarding the following sentence: "*Though neural networks (NNs) trained on relatively small datasets can generalize well, often significant trial and error is needed to select an architecture that does not overfit.*".
>
> We agree with the reviewer that, in heavily studied problems/datasets, such as image classification, the community has converged towards good all-around architectures (like ResNets). However, in domains where people are actively working to develop better NNs (such as for graph data, sets, point clouds, proteins, molecules, algorithmic reasoning) there is yet no consensus of what should be the correct architecture and simple choices can have a big effect on the achieved performance.
>
> The effect of simple architecture choices on generalization (in and out of distribution) has been documented in the literature, see for instance:
>
> * Xu et al. What Can Neural Networks Reason About. ICLR 2020 (see, e.g., Fig 4)
>
> * Veličković et al. Neural Execution of Graph Algorithms.  ICLR 2020 (notice the effect of the aggregation function on generalization)
>
> We will update the statement to clarify that we refer to problems that have not been intensely studied. We also remark that, by "does not overfit" we mean good-enough generalization and not "avoiding overfitting all together" (which we believe was the reviewer's interpretation).
>
> ### 3. Role and design of experiments
>
> Our experiments in the main text show how the same NN (e.g., MLP or CNN) trains differently for different target functions. These experiments are designed to match the theory by examining how the key quantities in the analysis (bias update, distance to initialization, and variance of bias) relate to the Lipschitz constant of the trained NN. Note that we control the NN's Lipschitz constant by increasing the Lipschitz constant of the target function (since the latter approximates the first when the loss is small).
>
> An additional experiment can be found in Section B. 5 of the supplementary material which shows the setting requested by the reviewer: different architectures solving the same task.
>
> It is also important to remark that our claims in the introduction (Section 1.2) and abstract provide a factual representation of our contributions.
>
> We believe that the reviewer has interpreted the open questions motivating this study, i.e., what is the connection between training and generalization, as a claim for contribution. Our study does not claim to have answered this question. We claim to provide mathematical evidence by proving that some quantities that are training-dependent can be used to deduce the NN complexity and generalization.
>
> ### 4. Contribution and novelty of experimental findings
>
> Our paper does not claim to have novel empirical findings w.r.t. how the complexity of the target function impacts the speed of convergence. Indeed, our paper cites previous studies (including the one pointed out by the reviewer) providing empirical evidence about these connections multiple times (lines 23-27 in Section 1 and line 287 in Section 6). On the contrary, these previous empirical works motivate the need for a theoretical analysis -- which is the main aim of our paper.
>
> Our contributions are that we prove specific mathematical relations between training (captured through the change of bias) and NN complexity (captured by the Lipschitz constant). The experiments examine exactly those connections.  We also do not prove something about the speed of convergence (which remains an open question).
>
> ### 5. Why do we focus on the bias
>
> We first clarify that we focus on the bias not by personal choice, but as an artifact of the mathematical analysis.
>
> It might at first be surprising that the bias is indicative of NN complexity. While we agree that the bias by itself does not reveal much, it turns out that the way the bias changes over successive SGD iterations reflects the operation of the entire NN. Indeed, as shown in the proof of Lemma 1, the expression of the gradient of $b_1$ with input $x$ is very close to the expression of the NN's Lipschitz constant at $x$. On the other hand,  the gradient of $W_1$ features additional terms that confound its meaning. Aiming to help future readers, we will explain these nuances in the relevant paragraph of Section 4 in the updated document.

---

### Decision · Program_Chairs · 2021-09-28

**Decision:**

Accept (Poster)

**Comment:**

Five knowledgeable reviewers recommend this paper for acceptance. Most concerns from the reviewers were addressed during the discussion and some reviewers increased their score after the discussion. One reviewer comments that this submission presents very concrete strong theoretical contributions, with a simple empirical study supporting them. I agree with the reviewers. Hence I recommend this paper for acceptance. For the preparation of the final manuscript, I encourage the authors to take the reviewers comments carefully into consideration. In particular: One reviewer is still concerned about the clarity of the paper. One reviewer mentions s/he doesn’t think the lack of experiments in the large scale setting is a primary weakness of this paper, but that having them would be great. Another reviewer still thinks that the quality of the paper could be greatly improved if it included some experiments (even if toy) to confirm the theoretical findings more directly and provided concrete suggestions. Another reviewer found that his/her questions were well addressed, but s/he still has some reservations about Assumption 1 and is not sure there is any chance it would hold in practice (although s/he does not see this as a major problem for the purposes of the theoretical analysis).

**Consistency Experiment:**

NeurIPS has a long history of experimentation. In 2014, NeurIPS ran an experiment in which 10% of submissions were reviewed by two independent committees to quantify the randomness in the review process. This year, we repeated a variant of this experiment to see how the quality of the review process has changed over time.  This paper was part of the experiment and was therefore assigned to two committees (consisting of reviewers, an Area Chair, and a Senior Area Chair) that reached independent decisions.  If both committees made the same recommendation, this recommendation was followed. If a single committee recommended acceptance, the paper was accepted (with the exception of a few cases in which the other committee identified what we considered a fatal flaw, e.g., an error in a key result).

Both committees reached the same decision: **Accept (Poster)**

The other committee assigned to the paper recommended **Accept (Poster)**.  You can find the other set of reviews, along with any follow up discussion with the authors here:
https://openreview.net/forum?id=RcjW7p7z8aJ